# Measurement report: The chemical composition and temporal variability of aerosol particles at Tuktoyaktuk, Canada during the Year of Polar Prediction Special Observing Period

John MacInnis[1], Jai Prakash Chaubey[1], Crystal Weagle[1,2], David Atkinson[3], Rachel Ying-Wen Chang[1]

[1]Department of Physics and Atmospheric Science, Dalhousie University, Halifax, B3H 4R2, Canada
[2]Department of Energy, Environmental and Chemical Engineering, Washington University in St. Louis, St. Louis, 63130-4899, United States of America
[3]Department of Geography, University of Victoria, Victoria, V8P 5C2, Canada

*Correspondence to*: Rachel Ying-Wen Chang (rachel.chang@dal.ca)

**Abstract.** The chemical composition, sources, and concentrations of aerosol particles vary on a seasonal basis in the Arctic. While existing research has focused on understanding the occurrence of aerosol particles during the Arctic winter and spring, less is known of their occurrence during the Arctic summer. In this study, atmospheric aerosol particle chemical composition and concentration were determined during July-September 2018 at Tuktoyaktuk, NT, Canada (69.4° N, 133.0° W) to coincide with the Year of Polar Prediction's 2nd Special Observing Period in the Arctic. The chemical composition of fine ($PM_{2.5}$) and coarse ($PM_{10-2.5}$) aerosol filter samples suggests the ocean, mineral/road dust, and combustion were sources of the sampled aerosol particles. Mass concentrations of $PM_2$ and $PM_{10}$, estimated from optical particle counter measurements, remained within a similar range during the study. However, elevated mass concentrations coincided with a festival in the community of Tuktoyaktuk, suggesting local human activity was an important source of aerosol particles. Mass concentrations of $PM_2$, which promote negative health effects in humans, were significantly lower at Tuktoyaktuk than the national air quality standard recommended by the Government of Canada. These measurements provide an important baseline to compare with future measurements associated with the assessment of aerosol chemistry and air quality in the Arctic.

## 1 Introduction

Aerosols are suspensions of liquid and solid particles in the atmosphere resulting from direct emissions from natural and anthropogenic sources and physical transformations, such as condensation and nucleation (Finlayson-Pitts and Pitts, 2000; Seinfeld and Pandis, 2016). The concentration, size distribution, and chemical composition of aerosol particles vary significantly in the atmosphere because their sources are diverse and ephemeral in nature (Finlayson-Pitts and Pitts, 2000; Seinfeld and Pandis, 2016). The role of aerosol particles in the atmosphere is complex: they serve as short-lived climate forcers (Willis et al., 2018), provide reactive surfaces for heterogeneous chemistry (Newberg et al., 2005), and are vectors for the

atmospheric transport (Wong et al., 2018) and post-depositional fate (MacInnis et al., 2019) of anthropogenic contaminants to and within remote Arctic environments, respectively.

In recent years, efforts have focused on understanding the source and chemical composition of aerosol particles in the Arctic, particularly within the context of climate change (Willis et al., 2018; Boy et al., 2019). Aerosol particle profiles vary on a seasonal basis near the surface in the Arctic. For example, the winter-spring period is characterized by high mass
concentrations of accumulation mode aerosol particles primarily from anthropogenic sources (i.e., Arctic haze), with a chemical composition dominated by sulphate ($SO_4^{2-}$) and black carbon (Sharma et al., 2017, 2019). However, natural sources of aerosol particles, such as mineral dust, have been identified in the Arctic during the winter-spring period (Dagsson-Waldhauserova et al., 2014, 2019; Bullard et al., 2016; Mackay and Burn 2005). The summer is a period characterized by low mass concentrations of aerosol particles primarily associated with the Aitken mode and natural sources (Chang et al., 2011;
Willis et al., 2018). Indeed, aerosol particles emitted from natural sources are important during the Arctic summer, however, it is anticipated that emissions from anthropogenic sources will increase as the Arctic responds to climate warming (Willis et al., 2018). An important consequence of climate warming is decreasing sea ice coverage, which promotes local warming through positive feedback albedo interactions, cloud and fog formation, the ocean-atmosphere exchange of aerosol particles and nucleation gases, and anthropogenic activities related to shipping and the exploration of minerals and fossil fuels (Willis
et al., 2018; Abbatt et al., 2019; Boy et al., 2019). Enhancements in summertime pollution promoted by anthropogenic activity may have an effect on the role of aerosol particles as climate forcers in the Arctic. For example, while it has been suggested that cooling is an ultimate effect of aerosol particles in the Arctic atmosphere, increasing emissions of black carbon from anthropogenic activities may contribute to Arctic warming through aerosol-radiation interactions (Willis et al., 2018). Changes in the chemical composition of aerosol particles may also affect their hygroscopicity and atmospheric fate as cloud
condensation nuclei, contributing to heating and cooling effects through aerosol-cloud interactions (Willis et al., 2018). The overall net impact of aerosol particles as short-lived climate forcers in the Arctic is unknown, however, increasing aerosol particle emissions are expected to have an impact on local air quality in the Arctic.

The size and chemical composition of aerosol particles has important implications for human health. In particular, aerosol particles with diameters less than or equal to 2.5 μm (referred to as $PM_{2.5}$) can accumulate in human lungs. These
particles have been associated with a number of diseases in humans, such as bronchitis, asthma, and cardiovascular disease (Dominici et al., 2006; Wu et al., 2018). In response to these concerns, the World Health Organization established a guideline for $PM_{2.5}$ of 25 μg m$^{-3}$ (24-hour average), which is suggested to be a level that minimizes risks to human health (World Health Organization, 2018). A similar guideline was established for $PM_{2.5}$ by the federal government of Canada (Canadian Ambient Air Quality Standard of 27 μg m$^{-3}$) (Environment and Climate Change Canada, 2018), although current research explores the
effects of aerosol particle exposure at lower concentrations (Brauer et al., 2019; Christidis et al., 2019; Pappin et al., 2019). For instance, several studies have noted that lower $PM_{2.5}$ mass concentrations (5 μg m$^{-3}$) have been associated with non-accidental death (Brauer et al., 2019; Pappin et al., 2019). Furthermore, the chemical components associated with $PM_{2.5}$ may be deleterious to human health, as it has been suggested that long-term exposure to various ions and metals in $PM_{2.5}$, including

$SO_4^{2-}$, nitrate ($NO_3^-$), potassium (K), iron (Fe), zinc (Zn), and silicon (Si) were associated with annual ischemic heart disease mortality in humans (Ostro et al., 2010).

Currently, the Arctic atmosphere is characterized by low concentrations of aerosol particles in the summer (Willis et al., 2018), however, it has been predicted that the chemical composition and magnitude of aerosol particle concentrations will change in the future as a result of climate change (Browse et al., 2013; Gilgen et al., 2018; Willis et al., 2018; Abbatt et al., 2019), which may have important implications on aerosol-radiation and aerosol-cloud interactions in the Arctic (Croft et al., 2019; Murray et al., 2021; Sanchez-Marroquin et al., 2020). In this study, the chemical composition and concentration of aerosol particles were determined in the western Canadian Arctic during the Year of Polar Prediction's (YOPP) 2nd Special Observing Period in 2018. The YOPP was an international collaboration led by the World Meteorological Organization designed to improve weather and sea ice forecasting and environmental prediction through the implementation of intensive and modelling campaigns in polar regions (https://www.polarprediction.net/). This work is a contribution to the YOPP, as the measurement of aerosol properties in the Canadian Arctic provides a baseline for future predictions of aerosol particle concentration and composition in this region.

## 2 Methods and data analysis

### 2.1 Study area and sample collection

Tuktoyaktuk, NT, Canada (69.45° N, 133.04° W), located at 5 m above sea level, is a hamlet bordering the Amundsen Gulf region of the eastern Beaufort Sea, and the latter is generally ice-covered during October-June (Herenz et al., 2018). It has a population of 982 (2018) and is accessible primarily via Inuvik-Tuktoyaktuk Highway (an unpaved, gravel road) or the local airport (Fig. 1). Tuktoyaktuk experiences a subarctic climate (Köppen climate classification) characterized by long, cold winters and brief, mild summers. The average annual temperature and total annual precipitation are less than 0 °C and 300 mm, respectively (Herenz et al., 2018).

Fine ($PM_{2.5}$) and coarse ($PM_{10-2.5}$) aerosol filter samples were collected from Tuktoyaktuk using an AirPhoton SS4i air sampler (Air Photon, Baltimore, USA, www.airphoton.com), as described previously (Snider et al., 2015), from 18 July to 12 September 2018. Briefly, air is drawn into the inlet of this assembly system by a vacuum pump and large particles (i.e., greater than 10 μm) are collected on a greased impaction plate. The sampled air then passes through a porous membrane (Nuclepore®, Whatman, 8 μm pore size) for the collection of $PM_{10-2.5}$, followed by a polytetrafluoroethylene (2 μm pore size) filter for the collection of $PM_{2.5}$. This assembly can accommodate eight filter units, which were gravimetrically weighed before and after the sampling period using a Sartorius Ultramicro Balance in a cleanroom with controlled relative humidity (32 ± 12 %) and temperature (20.9-21.0 °C) at Dalhousie University. Due to a file writing error during the study, the sample volume could not be determined and only the absolute mass collected on the filters is provided. We acknowledge this is a limitation of our study; however, the chemical composition of these samples remains of interest, as it is relevant to our understanding of the

atmospheric fate of aerosol particles, particularly during the Arctic summer in which existing measurements are scarce (Chang et al., 2011; Sierau et al., 2014; Tremblay et al., 2019).

Aerosol particles were sampled in situ using a GT-526S Handheld Particle Counter (Met One Instruments, Inc., Oregon, USA) from 24 July to 13 September 2018. The instrument was calibrated by the manufacturer before the study and sampled ambient air through an inlet line that was less than 1 m in length. This unit contains six channels that simultaneously measures

aerosol particles binned by diameter with lower limits of 0.3, 0.5, 1, 2, 5, and 10 μm, allowing the aerosol mass distribution to be characterized. Sample collection was performed every ten minutes at a flow rate of 2.8 L min$^{-1}$ during the study period.

The instruments in this study were collocated and mounted approximately 3.5 m above ground level at the Aurora College Community Centre (ACCC) in Tuktoyaktuk.

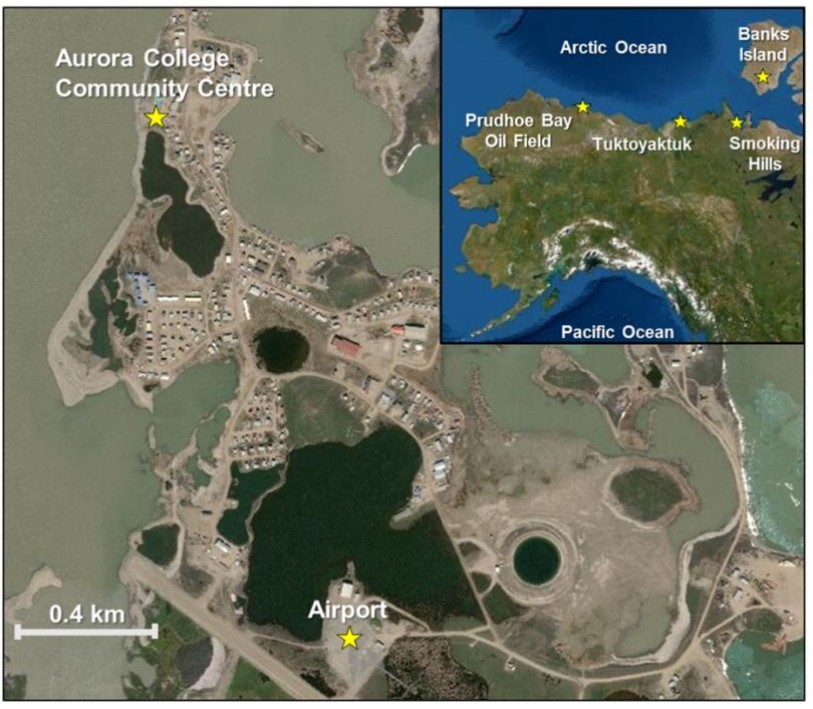


**Figure 1.** Location of the sampling site at the Aurora College Community Centre and the local airport in Tuktoyaktuk. Inset map shows the locations of Tuktoyaktuk and potential regional sources of aerosol particles, including the Smoking Hills, and Banks Island in Canada, Prudhoe Bay Oil Field in Alaska, and the Arctic and Pacific Oceans. Map source: Toporama (Natural Resources Canada). Note the geographic assignment of locations shown in the inset map is approximate.


The ACCC is located near residential and institutional buildings, a beach, and an unpaved road (all < 100 m from the site). Potential local sources of aerosol particles at the site may include dust (e.g., beach sand, road and mineral particles), as well

as marine and combustion aerosol particles resulting from natural and anthropogenic activities (e.g., sea spray and vehicle traffic). The ACCC could also be influenced by regional sources in northern Canada and the United States. For example, emissions from the ignition of lignite in the Smoking Hills and a migratory bird colony on Banks Island are possible sources of natural aerosol particles and gases, whereas emissions from the Prudhoe Bay Oil Field could be an anthropogenic source of aerosol particles and gases (Fig. 1). This site was selected for this study to represent a northern coastal community that was undergoing increased human activities (due to recent road access via the Trans-Canada Highway).

## 2.2 Laboratory analysis

Aerosol filter samples were extracted and analysed for ions and metals (Snider et al., 2015) at Dalhousie University in 2019. Before extraction, a ceramic blade was used to cut filters in half. One half of the filter was used for the analysis of water-soluble ions, including fluoride ($F^-$), chloride ($Cl^-$), bromide ($Br^-$), nitrite ($NO_2^-$), $NO_3^-$, $SO_4^{2-}$, phosphate ($PO_4^{3-}$), lithium ($Li^+$), sodium ($Na^+$), $K^+$, ammonium ($NH_4^+$), calcium ($Ca^{2+}$), and magnesium ($Mg^{2+}$), which were extracted using 3 mL of distilled water/isopropyl alcohol (4 %) and ultrasonication. Water-soluble ions were separated by a Dionex ICS-1000 ion chromatograph and analysed by conductivity detection. The other half of the filter was subjected to acid digestion using 3 mL of 10 % nitric acid ($HNO_3$) to extract metals, including Li, Mg, aluminum (Al), Fe, uranium (U), silver (Ag), barium (Ba), arsenic (As), Si, vanadium (V), chromium (Cr), manganese (Mn), cadmium (Cd), nickel (Ni), copper (Cu), Zn, antimony (Sb), cerium (Ce), lead (Pb), cobalt (Co), titanium (Ti), phosphorous (P), and selenium (Se). Metals were analysed by inductively coupled plasma mass spectrometry (ICP-MS, Thermo Scientific X-Series 2). Further details related to the extraction and analysis of filters can be found elsewhere (Snider et al., 2015).

## 2.3 QA/QC

Field blanks were used to investigate contamination introduced during the sampling and transport of filters in this study. Field blanks in this study were filters that were transported to the field site, but were not deployed, and returned to the laboratory for analysis concurrently with samples. Procedural blanks were used to investigate contamination introduced during extraction. These blanks consisted of the same analytical reagents used to extract filter samples, but without the aerosol filter matrix. Due to the brief sampling period in this study (i.e., estimated at 3 hours for each filter), the chemical mass of ions and metals in aerosol particle filter sample extracts were low and often estimated through extrapolation. We fully acknowledge that this practice introduces uncertainty into the quantitative assessment of ions in metals in aerosol particle filter samples and related analyses (Sect. 3.2). However, all reported masses are above the instrument detection limit (IDL). In addition, the chemical masses of ions and metals reported in this study were subjected to screening to account for their presence in procedural and field blanks (US Environmental Protection Agency, 2016), pursuant to protocols implemented by the US EPA: (1) if the analyte is detected in all field and procedural blanks (i.e., at masses equal to or greater than the IDL), then the detection limit is defined as the mean mass of analyte in blanks plus three-times the standard deviation (SD) of masses in the blanks; (2) if some, but not all, blanks contain analyte, then the detection limit is assigned to the highest mass observed in the blanks; or (3)

if the analyte is not detected in the blanks, then the detection limit is assigned to the IDL. The reported mass of ions and metals in filter samples are not blank subtracted (Supplement, Table S1). Although the magnitude of chemical mass reported in this study may carry uncertainty, the intention of this dataset is to provide a first assessment of the summertime chemical composition of aerosol particles in a relatively underreported region of the Canadian Arctic.

**2.4 Data analysis**

The absolute mass of ions and metals on filters was calculated by multiplying extract concentrations ($\mu$g mL$^{-1}$ or ng mL$^{-1}$) by the extraction volume (3 mL) and a factor of 2 to determine the mass of ions and metals on the entire filter. A mass reconstruction analysis was performed to estimate contributions by ions, metals, and particle-bound water to the total gravimetric mass, as described in the Supplement (Table S2). It was assumed that $NO_3^-$ and $SO_4^{2-}$ were neutralized by $NH_4^+$; however, we recognize that it is possible $NO_3^-$ and $SO_4^{2-}$ were associated with other species, depending on their source and
atmospheric fate (e.g., $NaNO_3$ produced from the acid displacement reaction with NaCl and $HNO_3$). Similarly, there is uncertainty related to the hypothesis that the source of Cl$^-$ in aerosol filter samples is limited to the ocean (i.e., NaCl component in Table S2), particularly due to the analytical challenges associated with the quantification of Na$^+$ in our samples. Nevertheless, we assumed that all Cl$^-$ in aerosol filter samples originated from marine aerosol particles for the purpose of this mass reconstruction estimate. Data from 27 August were not included in the calculation determining the average mass
reconstruction profile for PM$_{2.5}$ because the chemical mass sampled was larger than the gravimetric mass. Discrepancies between the total gravimetric mass and measured chemical mass of aerosol filter samples could be attributed to untargeted species (e.g., organics, other ions and metals, etc.), analytical uncertainties and/or losses of volatile species from filters during transport and laboratory analysis.

    Air mass back trajectories were calculated using the Hybrid Single Particle Lagrangian Integrated Trajectory Model
(National Oceanic and Atmospheric Administration, https://www.ready.noaa.gov/HYSPLIT_traj.php) to understand the source of ions and metals in aerosol filter samples. Air mass back trajectories were calculated over 120 hours using Global Data Assimilation System (GDAS) meteorology, setting the heights at the same location (end of parcel trajectory) to be 50, 200, and 400 m above ground level (Figs. S1 and S2). Available instrument log files suggest that sampling occurred once every eight days from 6:00 to 9:00 AM local time. Therefore, 9:00 AM was used as an end point for air mass back trajectory analysis.
Aerosol particle mass concentrations were estimated from measured aerosol particle number concentrations according to Eq. (1):

$$C_{Mass} = C_{Num} \cdot d_p^3 \cdot \rho_p \cdot \pi/6, \tag{1}$$

where $C_{Mass}$ is aerosol particle mass concentration ($\mu$g m$^{-3}$), $C_{Num}$ is the aerosol particle number concentration measured by the
particle counter (cm$^{-3}$), $d_p$ is the geometric mean diameter of aerosol particles in a given size bin (cm), and $\rho_p$ is aerosol particle

density, which was assumed to be 1.8 g cm$^{-3}$ (Sharma et al., 2017). PM$_2$ and PM$_{10}$ mass concentrations were calculated as the sum of mass concentrations in 0.3-2 μm and 0.3-10 μm size bins, respectively.

Meteorological data, including temperature, wind speed and direction, atmospheric pressure, and relative humidity was retrieved from historical climate archives (Environment and Climate Change Canada, 2020) for the Tuktoyaktuk airport during
the study (Fig. S3).

## 3 Results and discussion

### 3.1 Aerosol filter masses

The fine (PM$_{2.5}$, mean ± SD, 15 ± 9 μg, median 15 μg), coarse only (PM$_{10-2.5}$, 14 ± 4 μg, median 14 μg), and total coarse (PM$_{10}$,
29 ± 10 μg, median 26 μg) aerosol filter masses were similar during the study period, with notable variability (Fig. 2). For instance, the masses range from 2.6-31 μg, 7.3-22 μg, and 17-44 μg in PM$_{2.5}$, PM$_{10-2.5}$, and PM$_{10}$, respectively. Snider et al. (2016) also reported that masses of PM$_{2.5}$ (median 72 μg, lower-upper quintiles 42-131 μg) and PM$_{10-2.5}$ (median 90 μg, lower-upper quintiles 44-154 μg) were comparable in filter samples collected across a global network of sites (i.e., Surface PARTiculate mAtter Network, SPARTAN) using an AirPhoton sampler, although the exact distribution is site-specific. For
instance, comparable masses of PM$_{2.5}$ and PM$_{10-2.5}$ are not unexpected, considering coarse aerosol particle emissions are likely transient in nature (i.e., from local sources), and they may not have been sampled during the brief sampling period in this study. This is further supported by the mass distribution of fine and coarse aerosol particles measured by the particle counter, where the mass fraction of fine aerosol particles was occasionally higher than the mass fraction of coarse aerosol particles (Fig. S4). However, PM$_{10}$ masses in this study were always greater than PM$_{2.5}$ masses, as expected.
It is expected that the size distribution of aerosol particles (e.g., coarse mineral dust vs. fine combustion aerosol particles) and local meteorology could affect the magnitude of filter masses during the study. For example, it is possible that warmer temperatures during 26 July (Fig. S3) may have enhanced local emissions of coarse aerosol particles through heating and convection, contributing to a high PM$_{10-2.5}$ mass, while precipitation (i.e., drizzle, rain, and snow), which was observed at the airport before and during sampling events (Fig. S5), could have reduced atmospheric loads through the action of scavenging
aerosol particles and gases.

### 3.2 Chemical composition of aerosol filters

The detection frequencies for metals in aerosol filter samples were often low and/or variable, such as those observed for Ag, Ba, Cu, Sb, Ti, and Zn (Table S1).

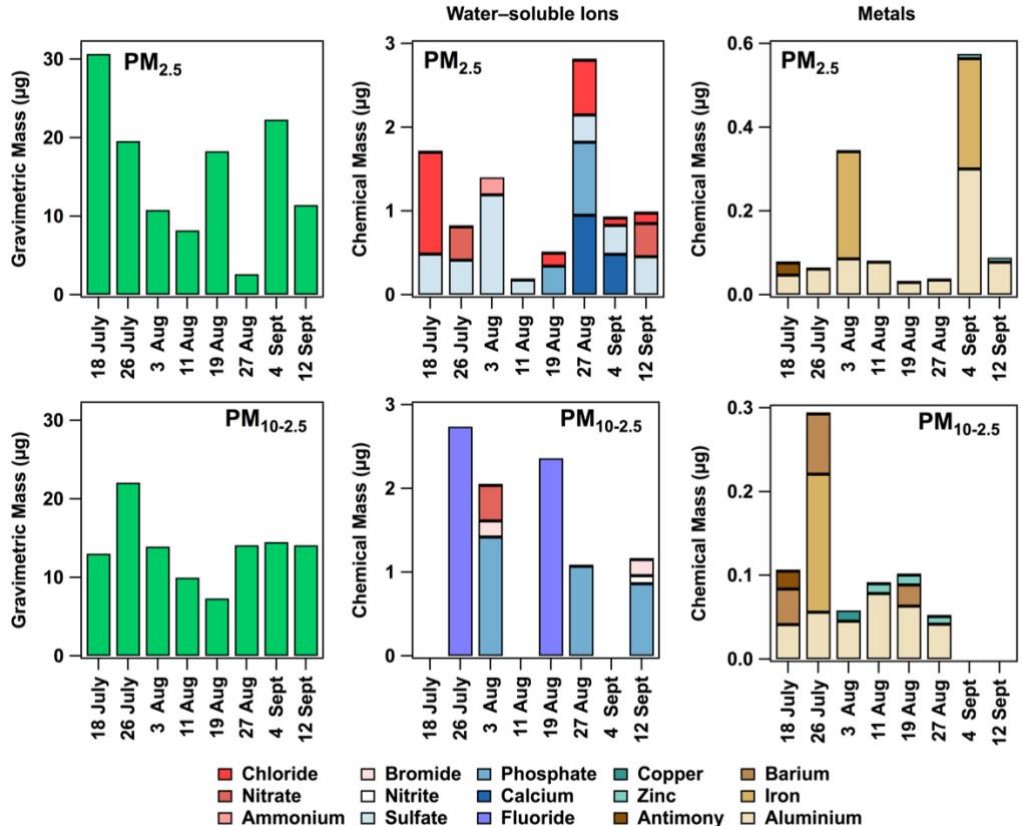

**Figure 2.** A summary of the gravimetric mass (left panels) and chemical mass of ions (middle panels) and metals (right panels) in fine (PM$_{2.5}$) and coarse (PM$_{10-2.5}$) aerosol filter samples (top and bottom row, respectively) from Tuktoyaktuk during July-September 2018.


In contrast, Al was detected in 100 % and 75 % of PM$_{2.5}$ and PM$_{10-2.5}$ samples, respectively. Interestingly, some metals were only detected in PM$_{2.5}$, while others were only detected in PM$_{10-2.5}$. For example, Ti was only detected in PM$_{2.5}$, whereas Cu, Ag, and Ba were only detected in PM$_{10-2.5}$, suggesting the latter may have been from local sources given their shorter atmospheric lifetime due to gravitational settling. Al and Fe were the metals measured in the highest quantities in aerosol filter

samples (Fig. 2).

The detection frequencies for water-soluble ions in aerosol filter samples were generally low and variable (Fig. 2). SO$_4^{2-}$ and Cl$^-$ were found in most PM$_{2.5}$ samples, with detection frequencies of 87 % and 62 %, respectively, whereas NO$_3^-$ was only detected in 25 % of PM$_{2.5}$ samples. Similar to metals, several water-soluble ions were only detected in select aerosol filter samples. For example, F$^-$, NO$_2^-$, and Br$^-$ were only detected in PM$_{10-2.5}$, suggesting a local source, whereas Cl$^-$, SO$_4^{2-}$, NH$_4^+$

and Ca$^{2+}$ were only detected in PM$_{2.5}$. The highest quantities of water-soluble ions in aerosol filter samples were observed for F$^-$, Ca$^{2+}$, PO$_4^{3-}$, and Cl$^-$ (Fig. 2).

Similar to other Arctic regions (Fig. 3), Al and Fe dominated aerosol filter samples at Tuktoyaktuk, which have been linked to mineral dust emissions (Liberda et al., 2015; Ferrero et al., 2019). In contrast, contributions from Ba and Sb to the total quantity of metals at Tuktoyaktuk were greater than other Arctic regions, whereas Ti contributions from other Arctic regions were greater than those at Tuktoyaktuk (Landsberger et al., 1990; Kadko et al., 2016; Conca et al., 2019). It is important to note that metal profiles in Landsberger et al. (1990) and Conca et al. (2019) are based on data collected during winter/spring periods, therefore seasonal differences in aerosol particles source in those studies may account for differences in composition profiles in comparison to Tuktoyaktuk (e.g., Arctic haze versus summertime sources). In addition, the studies compared in Fig. 3 do not always target the same ions and metals and/or face analytical challenges, preventing accurate reporting of data, which collectively could also contribute to chemical composition differences across sites. Ion composition profiles for aerosol filter samples at Tuktoyaktuk were notably different than other Arctic regions (Leaitch et al., 2018; Ferrero et al., 2019), with the exception of the $SO_4^{2-}$ in $PM_{2.5}$ at Tuktoyaktuk, whose composition profile was comparable to those found in aerosol particles from the Arctic Ocean and Ny-Ålesund (Fig. 3) (Ferrero et al., 2019). The ion composition profiles were dominated by $Na^+$ and $Cl^-$ at Ny-Ålesund and the Arctic Ocean, and by $SO_4^{2-}$ at Alert (Leaitch et al., 2018; Ferrero et al., 2019). $Na^+$ and $Cl^-$ have been linked to marine aerosol particle emissions, while $SO_4^{2-}$ has been linked to anthropogenic combustion and natural sources (Leaitch et al., 2018; Ferrero et al., 2019).

High detection frequencies of Al in $PM_{2.5}$ and $PM_{10-2.5}$ filter samples suggest that mineral dust was a consistent source of aerosol particles at Tuktoyaktuk. It is possible that local mineral/road dust emissions were sources of aerosol particles in this region, although the mass of Al was comparable in $PM_{2.5}$ and $PM_{10-2.5}$ filter samples, which may not be an outcome expected from local dust emissions if the size distribution of precursor particles (beach sand, soil, road dust) is coarse (i.e., higher mass expected in $PM_{10-2.5}$). Furthermore, it has been suggested that giant ($d_p > 40$ μm) and saltation mode ($3 \leq d_p \leq 6$ μm) mineral dust aerosol particles are primarily produced from soils (Saltzman, 2013). This could suggest that in addition to local sources, the presence of Al in filter samples was attributed to mineral dust emissions and atmospheric transport from other, high latitude regions within the Arctic (Crocchianti et al., 2021; Mackay and Burn 2005). For example, it is possible that aerosol filter samples on 18 and 26 July (Fig. S1) were influenced by active dust sources near the North Slope of Alaska (https://maps.unccd.int/sds/).

It is noteworthy that other metals characteristic of mineral dust, such as Fe and Ti, were detected less frequently in filter samples (Table S1), suggesting they had different sources. Interestingly, the highest mass of Al, Ti, Fe, and Zn in $PM_{2.5}$ filter samples was observed on 4 September, and air masses during this period originated from the north and travelled in north westerly directions, over the ocean. Other air mass trajectories originating from west and north westerly directions (i.e., Alaska and Russia) were observed on 18, 26 July and 19 August (Figs. S1 and S2), and filter samples during these periods contained Ba, Ag, and Sb. It is possible that these air masses were influenced by emissions from the Prudhoe Bay Oil Field and mining activities in Alaska, Russia, and Canada during these periods (Alaska Miners Association, 2020; Government of Canada, 2018; European Environment Agency, 2017). However, local emissions from combustion and natural or anthropogenic dust (e.g.,

road dust containing tire wear and mineral/soil particles) (Snider et al., 2016; Crocchianti et al., 2021; Mackay and Burn 2005) cannot be precluded as sources of Al, Fe, Ti, Zn, Ba, Ag, and Sb in filter samples.

It is interesting that $Cl^-$ was only detected in $PM_{2.5}$ while $Br^-$ was only detected in $PM_{10-2.5}$ filter samples since $Cl^-$ and $Br^-$ have been measured in seawater from the Canadian Arctic Archipelago (Xu et al., 2016). Although the ACCC is a costal site, surface meteorology records from the airport (Fig. S3) indicated that local wind speeds were often below 4 m s$^{-1}$ (e.g., 26

July, 19 and 27 August, 12 September), which has been suggested as a threshold wind speed for whitecap formation (O'Dowd and de Leeuw, 2007). However, a marine influence was expected during 3 and 11 August and 4 September, since wind speeds were greater than 4 m s$^{-1}$ and originated from north westerly and easterly directions (Fig. S3). If the presence of $Br^-$ in $PM_{10-2.5}$ filter samples was attributed to marine aerosol particles, then $Cl^-$ should have also been present based on the molar composition of $Cl^-$ and $Br^-$ in seawater from the Canadian Arctic Archipelago (Xu et al., 2016). On the other hand, it is important to consider

that the absence of $Br^-$ in $PM_{2.5}$ filter samples could be related to analytical challenges because the mass of $Br^-$ expected in $PM_{2.5}$ filter samples, based on the molar composition of $Cl^-$ and $Br^-$ in seawater from the Canadian Arctic Archipelago (Xu et al., 2016), is below the detection limit in this study.

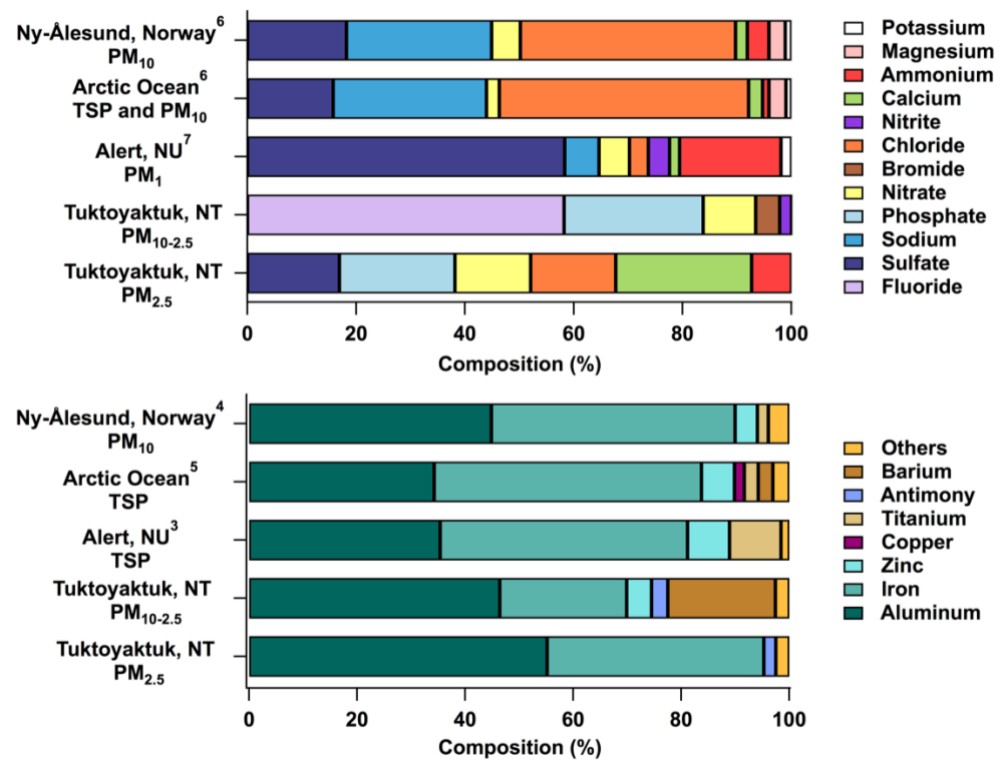

**Figure 3.** Composition profiles of ions (top panel) and metals (bottom panel) in aerosol filter samples at Tuktoyaktuk and other Arctic regions. Average composition profiles are shown for aerosol filter samples at Tuktoyaktuk. Only ions and metals that are equal to or greater than the detection limit in this study are included in this figure. Several elements in Landsberger et

al. (1990)[3] (Ca, Cl, Na), Conca et al. (2019)[4] (Mg, K, Na, Ca), and Kadko et al. (2016)[5] (Na, Mg) were detected in aerosol particles, however, they are not included in this figure to enable comparisons with trace metal composition profiles. Data shown for Leaitch et al. (2018)[7], Kadko et al. (2016)[5], Ferrero et al. (2019)[6], Landsberger et al. (1990)[3], and Conca et al. (2019)[4] correspond to sampling periods during 8 August 2014, 14 August 2011, 20 June to 12 August 2011-2012, 7-18 March 1985, and March to September 2010-2013, respectively.

It is unlikely that the absence of $Cl^-$ in $PM_{10-2.5}$ filter samples can be explained by acid displacement reactions, considering it has been suggested that $Cl^-$ depletion decreases with increasing aerosol particle size (Yao et al., 2003). Thus, these observations suggest that $Cl^-$ and $Br^-$ were from different sources. In addition to marine sources, $Cl^-$ and $Br^-$ can originate from biomass burning. Keene et al. (2006) identified hydrochloric acid (HCl), chlorine ($Cl_2$), hypochlorous acid (HOCl), bromine ($Br_2$), and hypobromous acid (HOBr) as products of biomass burning, which could have been the source of either $Cl^-$ measured in the fine mode or $Br^-$ measured in the coarse mode at Tuktoyaktuk. While we are unable to confirm this source in our study, it is conceivable that biomass burning in northern Canada was a possible source of aerosol particles at Tuktoyaktuk (e.g., 27 August, Fig. S2).

It is challenging to identify sources of ions and metals in aerosol filter samples, especially when they are emitted from various sources in the environment (Jayarathne et al., 2014; Leaitch et al., 2018; Willis et al., 2018). For instance, it is likely that ions and metals originated from continental and marine sources in this study, pursuant to air mass back trajectory analysis during the study period (Figs. S1 and S2). One strategy that can be used to better constrain continental and marine sources is to compare known ratios of ions and metals in soils and seawater to those found in aerosol filter samples. As an example, the molar ratios of Fe/Al and $Cl^-/SO_4^{2-}$ in $PM_{2.5}$ from Tuktoyaktuk are compared to those found in Icelandic soils (Đorđević et al., 2019) and seawater from the Canadian Arctic Archipelago (Xu et al., 2016), respectively. The molar ratio of Fe/Al in Icelandic soils is 0.9, and this ratio is within the range of those observed in aerosol filter samples from Tuktoyaktuk (0.4-1.4). This result further supports the hypothesis that Fe and Al could originate from mineral dust. In contrast, the molar ratio of $Cl^-/SO_4^{2-}$ in Arctic seawater is 27, which is higher than those in $PM_{2.5}$ from Tuktoyaktuk (0.7-6.8). Lower molar ratios of $Cl^-/SO_4^{2-}$ in aerosol filter samples could be attributed to non-oceanic sources of $SO_4^{2-}$ (i.e., natural and anthropogenic combustion sources). For example, $SO_4^{2-}$ in $PM_{2.5}$ at Tuktoyaktuk may have originated from natural sources, such as the biogenic emission and subsequent oxidation of dimethyl sulphide from the ocean (Bates et al., 1987). In addition, sulfur emissions from the Prudhoe Oil Fields and the ignition of lignite in the Smoking Hills (Radke and Hobbs, 1989) were likely sources of $SO_4^{2-}$ in $PM_{2.5}$ at Tuktoyaktuk, according to air mass back trajectories (Figs. S1 and S2). Another source of $SO_4^{2-}$ in $PM_{2.5}$ at Tuktoyaktuk may include anthropogenic emissions from the combustion of fossil fuels (e.g., vehicles, aircraft, boats, etc.) (Leaitch et al., 2018; Willis et al., 2018). Other ions characteristic of combustion were also identified in aerosol filter samples from Tuktoyaktuk, such as $NO_3^-$ and $NH_4^+$, possibly from the emission and oxidation of nitrogen oxides ($NO_x$) and emissions of ammonia during fossil fuel combustion from local, regional (e.g., Prudhoe Oil Fields), and long-range sources. However, ammonia emissions in the Arctic have also been associated with natural sources, such as soil (Wentworth et al., 2016) and guano (Croft et al.,

2016; Wentworth et al., 2016), which could account for $NH_4^+$ in aerosol filter samples at Tuktoyaktuk, particularly on 3 August because $NH_4^+$ was detected in air masses that travelled near a bird colony on Banks Island before arriving at the ACCC (Fig. S1). It is important to consider that lower molar ratios of $Cl^-/SO_4^{2-}$ in $PM_{2.5}$ filter samples from Tuktoyaktuk could also be

attributed to $Cl^-$ depletion from marine aerosol via acid displacement (Newberg et al., 2005). For example, it is possible that $HNO_3$ and sulfuric acid ($H_2SO_4$) displaced $Cl^-$ in $PM_{2.5}$ from Tuktoyaktuk (Laskin et al., 2012), particularly since $NO_3^-$ and $SO_4^{2-}$ were detected in aerosol filter samples that were devoid of $Cl^-$ (i.e., 26 July-11 August). Although the organic composition of aerosol filter samples was not characterized in this work, it is important to consider that organic acids (Laskin et al., 2012) may have also contributed to $Cl^-$ depletion in aerosol filter samples.

315        A mass reconstruction analysis was conducted to estimate contributions from ions and metals to the total gravimetric mass (Bari and Kindzierski, 2017). The average mass reconstruction profile for $PM_{2.5}$ indicates that contributions from marine and mineral dust sources collectively accounted for 43 % of the total known chemical mass (i.e., NaCl and mineral dust components), while contributions from $(NH_4)_2SO_4$ and $NH_4NO_3$ collectively accounted for 31 % of the known chemical mass (Fig. 4). Lower contributions were observed for metals and particle-bound water, collectively accounting for 6 % of the total

known mass in $PM_{2.5}$ filter samples. In contrast, the average mass reconstruction profile for $PM_{10-2.5}$ was dominated by $NH_4NO_3$ and unclassified inorganic ions (i.e., combined masses of $F^-$, $Br^-$, $PO_4^{3-}$, $NO_2^-$, Table S2), which collectively accounted for 93 % of the total chemical mass. This analysis suggests that marine, mineral dust, and combustion sources were important in this region; however, the source of $PO_4^{3-}$ and $F^-$, which were major components of the unclassified inorganic fraction of aerosol particle filter samples, could not be identified.


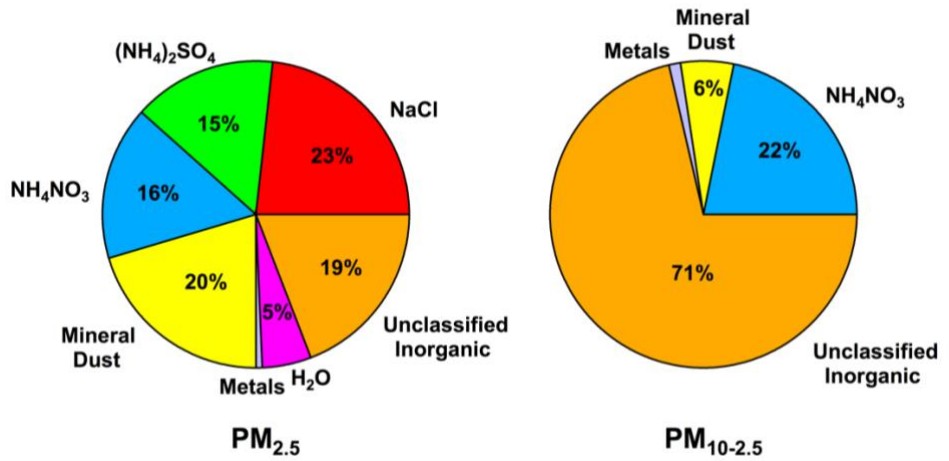

**Figure 4.** Average mass reconstruction profiles of fine and coarse aerosol filter samples at Tuktoyaktuk during July-September 2018. Several ions measured in this study ($F^-$, $Br^-$, $PO_4^{3-}$, $NO_2^-$) are collectively included here as unclassified inorganic components due to uncertainties associated with their sources in the environment.


The sources of these ions are interesting because $PO_4^{3-}$ and $F^-$ have not been previously identified in aerosol particles from the Arctic (Fig. 3). We acknowledge that it is possible that the detection of $F^-$ could be related to laboratory contamination, however, the reported masses in this study are equal to or greater than the detection limit (i.e., mean + 3SD of blank masses). It is challenging to assign sources to $PO_4^{3-}$ and $F^-$ because they are released from multiple sources in the environment, such as

emissions from volcanos, mineral dust, combustion, and marine aerosol particles (Jayarathne et al., 2014; Vet et al., 2014). It is also challenging to assign sources to $F^-$ because one of its gas phase precursors (i.e., hydrofluoric acid, HF), emitted during smelting activity, coal combustion, and volcanic activity (Jayarathne et al., 2014) may partition to aerosol particles in the Arctic atmosphere. While there is uncertainty associated with the source of $PO_4^{3-}$ and $F^-$ in aerosol filter samples, the back trajectories suggest that air masses containing $PO_4^{3-}$ travelled primarily over the ocean, whereas air masses containing $F^-$

travelled over the ocean and land (Figs. S1 and S2). These observations suggest that marine, mineral, and combustion aerosol particles are possible sources of $PO_4^{3-}$ and $F^-$ at Tuktoyaktuk.

In addition to uncertainties associated with the unclassified inorganic fraction, a substantial fraction of mass on aerosol filter samples could not be identified. For instance, the average chemical mass only accounted for 10 % and 12 % of the total gravimetric mass in $PM_{2.5}$ and $PM_{10-2.5}$ filter samples, respectively (Fig. S6). Discrepancies between the total gravimetric mass

and chemical mass of aerosol filter samples could be attributed to analytical uncertainties, the loss or gain of volatile species from filters after sampling (Saltzman, 2013), and/or contributions from untargeted chemical components. For example, it is possible that other inorganic ions and metals, organic material, and black carbon were components of aerosol particles at Tuktoyaktuk, pursuant to the chemical composition of aerosol particles in other Arctic regions (Kadko et al., 2016; Leaitch et al., 2018; Conca et al., 2019; Ferrero et al., 2019; Sharma et al., 2019). Despite these uncertainties, our results indicate that

there are substantial, uncharacterized chemical components that could not be identified in aerosol filter samples at Tuktoyaktuk.

The summertime composition profiles at Tuktoyaktuk may provide insights into the future chemical composition of aerosol particles in the Arctic. Our analysis indicates that mineral dust, marine, and combustion sources are important during the Arctic summer, particularly in the absence of snow cover and sea ice coverage. It is expected that emissions of these aerosol

particles will increase as the Arctic responds to climate warming in the future. For example, it is expected that climate warming will enhance the melting of snowpacks and sea ice within the Arctic, promoting exposed landscapes and oceans, and ship traffic (Willis et al., 2018). Our analysis also indicates that there are significant, unknown components of aerosol filter samples during the summer of 2018 at Tuktoyaktuk, which may influence the chemical properties of aerosol particles and their role in the Arctic troposphere (e.g., cloud condensation nuclei, radiative properties) (Martin et al., 2011; Herenz et al., 2018; Willis

et al., 2018; Abbatt et al., 2019). For example, the unidentified components (e.g., organic components) are likely less hygroscopic than the soluble inorganics identified and would reduce the cloud condensation nuclei activity of the aerosol particles. This highlights the importance of improving understanding of the chemical composition of aerosol particles in the Arctic. It is important to emphasize that these results only provide a snapshot of the aerosol particles at Tuktoyaktuk and their representativeness is unknown.

**3.3 Size distribution, temporal variability, and health implications of aerosol particles**

The average number size distributions of particles larger than 0.3 μm were similar throughout the study, with particle number concentrations highest in the 0.3-0.5 μm bin (Fig. S7). The mass size distributions also remained similar throughout the study, with mass concentrations dominated by the 2-5 μm aerosol particles (Fig. S7).

Total aerosol particle number concentrations of particles larger than 0.3 μm ($N_d$) in this study were low (1-hour average, 370 mean ± SD, $6 \pm 8$ cm$^{-3}$, median 3 cm$^{-3}$). These relatively low concentrations are consistent with other observations during the Arctic summer (Willis et al., 2018) and may have been attributed to enhanced aerosol particle and nucleation gas scavenging by precipitation (Croft et al., 2016). Aerosol particle mass concentrations for $PM_{2.5}$ and $PM_{10}$ were similar in regions of the Northwest Territories, Yukon Territory, and Nunavut (Canadian Council of Ministers of the Environment, 2019) but were higher than those observed for $PM_2$ and $PM_{10}$ at Tuktoyaktuk during July-September 2018 (Table 1).


**Table 1.** An overview of mass concentrations for $PM_2$, $PM_{2.5}$, and $PM_{10}$ at Tuktoyaktuk and other regions in Canada (Canadian Council of Ministers of the Environment, 2019) from 26 July to 13 September 2018. Population data for 2018 was retrieved from the Northwest Territories Bureau of Statistics (NWT Bureau of Statistics, 2020), the Government of Nunavut (Government of Nunavut, 2019), and the Yukon Bureau of Statistics (Yukon Bureau of Statistics, 2019).

| Location | Tuktoyaktuk, NT 69º N | | Inuvik, NT 68º N | | Norman Wells, NT 65º N | | Iqaluit, NU 64º N | Yellowknife, NT 62º N | | Whitehorse, YT 61º N |
|---|---|---|---|---|---|---|---|---|---|---|
| Population | 982 | | 3,536 | | 818 | | 38,139 | 20,607 | | 40,643 |
| μg m$^{-3}$ | PM$_2$ | PM$_{10}$ | PM$_{2.5}$ | PM$_{10}$ | PM$_{2.5}$ | PM$_{10}$ | PM$_{2.5}$ | PM$_{2.5}$ | PM$_{10}$ | PM$_{2.5}$ |
| Mean | 0.9 | 2.3 | 2.6 | 13 | 3.7 | 11 | 4.3 | 2.6 | 16 | 2.4 |
| Median | 0.6 | 1.3 | 2.0 | 6 | 3.0 | 7 | 4.0 | 0.0 | 13 | 2.0 |
| Max | 6.9 | 48 | 19 | 133 | 22 | 122 | 15 | 58 | 97 | 48 |


Although some measurements at Tuktoyaktuk are not directly comparable to those of other regions in Canada (i.e., $PM_2$ vs. $PM_{2.5}$), it is possible that the difference in the magnitude of aerosol particle mass concentrations could be attributed to differences in regional climate (i.e., precipitation) and aerosol particle sources. For instance, it is likely that Tuktoyaktuk is influenced by marine air masses to a greater extent than other sites listed in Table 1, which are generally located inland. 385 Differences in regional populations/human activities could further promote differences in aerosol particle concentrations (e.g., enhanced vehicle traffic). It is interesting to note in Table 1 that relatively high aerosol particle mass concentrations were observed at Norman Wells despite its low population, however, this may be attributed to major crude oil and natural gas production in that region (Canada Energy Sector, 2020).


The difference in the magnitude of aerosol particle mass concentrations may also be related to differences in sampling techniques. For example, one method used by the National Air Pollutant Surveillance program to determine mass concentrations is by filtration and beta attenuation (Canadian Council of Ministers of the Environment, 2019) whereas the method used here relies on aerosol particle number concentrations and estimations of aerosol particle density (Eq. 1). Despite

these differences, these data indicate that aerosol particle concentrations were lower during the summer of 2018 at Tuktoyaktuk than other regions in northern Canada.

A summary of the continuous aerosol particle mass concentrations during July-September 2018, calculated from the particle counter, is presented in Figs. 5 and 6. In general, the magnitude of mass concentrations remained within a similar range during July-September 2018, although there were notable increases in mass concentrations during 28-30 July, 3-6

August, and 22-24 August (Fig. 5). For example, the average $PM_{10}$ concentration was 2.3 $\mu g\ m^{-3}$ over the study period, whereas higher $PM_{10}$ concentrations occurred on 28 July (35 $\mu g\ m^{-3}$), 4 August (48 $\mu g\ m^{-3}$), and 24 August (40 $\mu g\ m^{-3}$). Although we are unable to confirm the source of aerosol particle emissions during these periods, it is conceivable that elevated mass concentrations could be attributed to local aerosol particle emissions from human activities at Tuktoyaktuk.

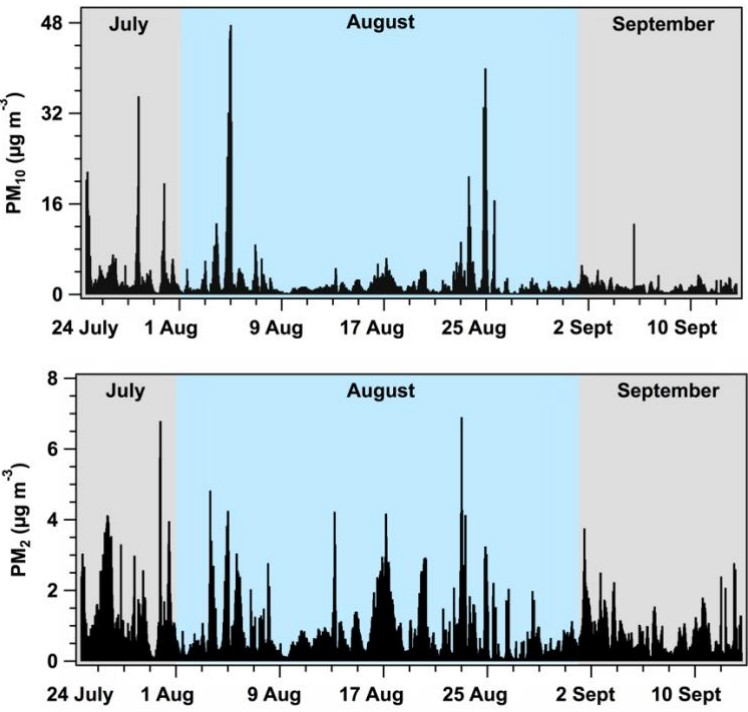

**Figure 5.** Mass concentration profiles for $PM_2$ and $PM_{10}$ at Tuktoyaktuk during July-September 2018. The data are presented as 1-hour averages (Mountain Daylight Time), and each monthly period is demarcated by colour.

For instance, elevated PM$_{10}$ concentrations on 28 July may have been related to human activities that proceeded during the Annual Land of the Pingos Musical Festival (e.g., barbequing, increased vehicle traffic), which occurred during 27-29 July at Tuktoyaktuk. Similar human activities could also provide an explanation for elevated mass concentrations on 3-6 August, considering this period coincided with an extended civic holiday weekend.

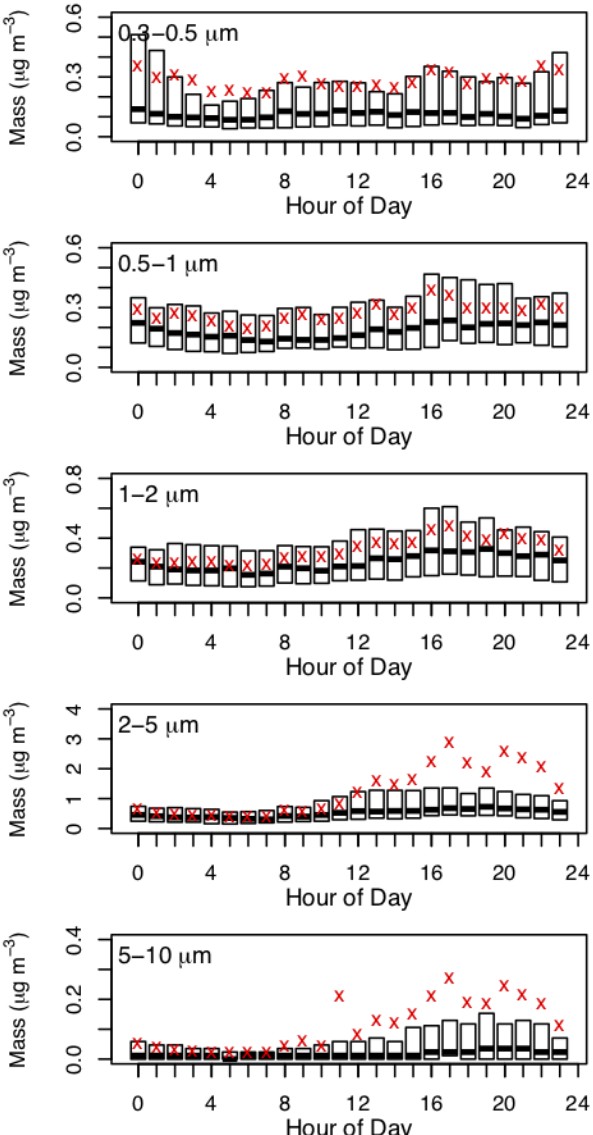

**Figure 6.** Box plot of diurnal aerosol particle mass concentrations (μg m$^{-3}$) at Tuktoyaktuk during 24 July to 13 September 2018 (Mountain Daylight Time). Average concentrations are denoted by red markers, median concentrations correspond to black lines within each box, and the lower and upper bounds of each box are the 25$^{th}$ and 75$^{th}$ percentiles, respectively.

Aerosol particle mass concentrations did not exhibit notable diurnality during the study (Fig. 6). Average mass concentrations were typically higher than median mass concentrations and exhibited notable variability in the 2-5 and 5-10 μm size bins, which are likely driven by enhanced aerosol particle emissions from local human activities at Tuktoyaktuk, as discussed previously (i.e., festival and weekend activities).

To assess the impact of fine aerosol particle emissions on local air quality, the mass concentration of $PM_2$ at Tuktoyaktuk was compared to the Canadian Ambient Air Quality Standard. The concentration of $PM_2$ was low at Tuktoyaktuk during July-September 2018, with 24-hour average concentrations ranging from 0.2-3 μg m$^{-3}$. These mass concentrations were very low in Tuktoyaktuk compared to the national $PM_{2.5}$ standard (27 μg m$^{-3}$).

## 4 Conclusions

The chemical composition of aerosol filter samples and concentration of aerosol particles from Tuktoyaktuk were determined during July-September 2018. Although our analysis could not identify distinct sources, the results suggest that this moderately-sized community in the Canadian north was influenced by a wide range of aerosol particle sources with complex processes. The observed aerosol particles were likely derived from local natural sources like marine and mineral dust and anthropogenic sources like the combustion of fossil fuels and road dust, while emissions from the Prudhoe Oil Field, Smoking Hills, bird colonies on Banks Island, mining activities in northern Canada, Russia, and Alaska, and mineral dust from active source regions in the Arctic are possible regional sources of aerosol particles, pursuant to air mass back trajectory analysis (Figs. S1 and S2). We hypothesize that precipitation reduced atmospheric loads of aerosol particles and gases during the study, which is expected to affect the magnitude of the gravimetric mass and chemical composition of aerosol filters and at Tuktoyaktuk, and air temperature may have enhanced local emissions of coarse aerosol particles through daytime heating and convection. Our analysis indicates that there were significant, unknown components identified in aerosol filter samples during the summer of 2018 at Tuktoyaktuk, which may influence the atmospheric fate of aerosol particles in the Arctic troposphere. While the mass concentrations of $PM_2$ were found to be significantly lower at Tuktoyaktuk compared to the Canadian Ambient Air Quality Standard, it is likely that their concentrations will increase in the future due to climate change, which is expected to promote increases in ship and air traffic in the Arctic as well as the number of ice-free days and natural emissions from open waters. Although these measurements only represent a snapshot of the aerosol particles at Tuktoyaktuk, they can nevertheless provide insights into the chemistry and concentration of aerosol particle samples, which can be used in the future to assess aerosol particle chemistry and air quality in the Canadian Arctic. Future work should focus on constraining possible sources of aerosol particles, such as acquiring time-resolved chemical mass spectra data and performing factor analysis (e.g., positive matrix factorization) and/or analysing the chemical composition of local soils.

*Data availability*. Quality-controlled aerosol filter chemistry and aerosol particle number concentrations are available through the Federated Research Data Repository (https://doi.org/10.20383/101.0269 and https://doi.org/10.20383/101.0278).

*Author contribution*. RC and DA were responsible for funding acquisition and conceptualized the research. JC and DA were responsible for data acquisition. JC and CW contributed to data curation. The formal analysis and visualization of data was performed by JM. JM and RC wrote the manuscript, with editorial feedback from all co-authors.

*Competing interests*. The authors declare that they have no conflict of interest.

*Acknowledgements*. The authors thank George Hibbs for providing access to the sampling site, Nicole Chisholm for weighing
filters, Brenna Walsh for helpful discussions, and Randall Martin and Jong Sung Kim for providing resources during the deployment. This work was funded by the Marine Environmental Observation, Prediction, and Response Network (MEOPAR), Polar Knowledge Canada [NST-1718-0001] and the Canada Research Chair program [CRC-2013-00056]. This is a contribution to the Year of Polar Prediction (YOPP), a flagship activity of the Polar Prediction Project (PPP), initiated by the World Weather Research Programme (WWRP) of the World Meteorological Organisation (WMO). We acknowledge the
WMO WWRP for its role in coordinating this international research activity.

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
