# Peer review of "Measurement report: The chemical composition and temporal variability of aerosol particles at Tuktoyaktuk, Canada during the Year of Polar Prediction Special Observing Period"

_Atmospheric Chemistry and Physics, 2021_

## Referee Comment (RC2)

**Reviewer's report on the manuscript by MacInnis et al. "Measurement report: The chemical composition and temporal variability of aerosol particles at Tuktoyaktuk, Canada during the Year of Polar Prediction Special Observing Period", Atmospheric Chemistry and Physics, Manuscript ID: acp-2021-262**

The manuscript presents a report on the measurements carried out at a small Inuit community located near the Mackenzie River delta during summertime in 2018. The measurements include filter samples and continuous sampling using a particle counter. Analysis were carried out in an effort to characterise particulate matter compositions observed at this sites during the summer period. The manuscript is written as a measurement report, due to, I gather, some issues with the measurements (e.g., missing sample volume information, insufficient sampling period due to low concentrations) which resulted in certain limitations of the study. I do feel that the authors could extend the analysis a bit more to improve the interpretation of the measurement results and gain better insight into the observation at this remote Arctic site. My comments and suggestions are provided below.

2. Method and data analysis

2.1 Study area and sample collection

Could the authors provide some background on why this particular site was chosen and a description of the site and the area in terms of potential sources of aerosols?

The authors acknowledge that the sample volume could not be determined due to a file writing error so that air concentrations cannot be discerned from the filter samples. Were airflow rate and length of sampling controlled?

Is it correct that the particle counter has a lower size cut at 300 nm (so that particles smaller than 300nm are not measured by this instrument)? This seems to limit the ability to characterize aerosol size spectrum from this study as it misses Aitken mode almost entirely.

Were the inlets (filter sampling and particle counter) collocated?

2.3 QA/QC

Line 121: "the unanticipated, brief sampling period" – what do you mean?

Line 121 – 124: How should this low mass issue be taken into consideration with regard to the results shown in Figure 2?

2.4 Data analysis

Although the authors did include statistical summaries of meteorological observations (from the Tuktoyaktuk airport) during the field measurement period (Figure S1), it would be much more useful for data interpretation to plot the time series (e.g., wind speed/direction, temperature, humidity) as well.

3. Results and discussion

3.1 Aerosol filter masses

Line 167 – 169: The authors seems to suggest that the $PM_{2.5}$ and $PM_{10\text{-}2.5}$ masses based on filter measurements at this site were comparable (in terms of means and ranges; what about median?) and that the comparable masses between $PM_{2.5}$ and $PM_{10\text{-}2.5}$ were also shown from the global SPARTAN network sites (using the same instrument and analysis method). What does this imply? Is this corroborated by the mass estimates based on the particle counter measurements?

Line 169 – 171: Could the authors elaborate on this a bit more? How are the meteorological conditions related to the observed PM mass levels and how are PM levels affected by local and distant sources?

3.2 Chemical composition of aerosol filters

Figure 2 shows both the gravimetric masses and chemical masses from each of the filter samples. It would be interesting to see the mass differentials between the gravimetric mass and the total chemical mass from each of the samples to get an idea on how much of the PM mass is explained by the speciation and how much is unexplained (given that the analysis covers inorganic ions and metals but not organics). Perhaps this will provide some additional information for source identification under different conditions.

Is there any correlation between the variation in gravimetric masses amongst the filter samples and the variation in PM concentrations derived from the aerosol counter measurement?

Is sodium not analysed? Is sulfate shown including sea-salt sulfate?

Line 193 – 195: It might be good to rephrase this, as the only common feature shared in chemical composition of metals between Tuktoyaktuk and other Arctic sites shown in Figure 3 is the dominance of Al and Fe.

Figure 3: Are chemical composition profiles from other Arctic sites shown here based on summertime measurements also? If not, how might seasonal variability affect the comparison here? Also for the comparison do all sites carry out analysis for the same suite of ions and metals? For example, the Tuktoyaktuk profiles do not include sodium; does it mean that sodium is not present or just not analysed?

Line 211 – 214: Longer back trajectories are needed to better discern air mass origin (or influence) in the Arctic summertime, given that air mass tends to resides within the Arctic region for a long time (up to 2 weeks) in summertime (Stohl, 2006, JGR).

Line 235 – 240: Do the author imply that the $Cl^-$ and $Br^-$ detected in $PM_{2.5}$ and $PM_{10\text{-}2.5}$ samples, respectively could be of biomass burning origin? It would have been possible for $Cl^-$ in $PM_{2.5}$ but one would not expect coarse particles to be transported from a long distance. It is still surprising not to see $PM_{10\text{-}2.5}$ sea salt at this coastal site.

It would be helpful to include a description of the local and regional sources (natural and anthropogenic). The influence of Smoking Hills emissions and Prudhoe oil fields could be discerned from trajectory analysis. For example, the August 3 sample could be influenced by sulfur emissions from Smoking Hills (based on the trajectory shown in Figure S2).

3.3 Size distribution, temporal variability, and health implications of aerosol particles

It should be noted that the aerosol number size distribution based on this measurement is incomplete as the measurement is missing Aitken mode particles almost entirely (with the lowest size cut at 300 nm).

Line 318: What do you mean by number size distribution being consistent with Herenz et al. (2018)? Their number size distributions show highest mode at ~40 – 50 nm under polluted conditions and just below 200 nm under clean conditions (their Figure 5). Those measurements were conducted during spring-to-summer transition period while this study is during summer period. One would expect to see quite significant differences in aerosol size distribution and chemical composition between these two different periods. Would this not be the case?

Table 1: Please clarify on $PM_{2.5}$ and $PM_{10}$ measurements at the NAPS sites. They may be using different instrument/technique than that used in this study.

Line 326, Line 339, and Line 343: It may be more appropriate not to use the term "discrepancy" (or "discrepancies") here. The differences are expected between these different northern sites, due to, as the authors pointed out, the differences in geographical locations, local and regional sources, etc.

Line 343 – 344: It may be better to say "… concentrations were lower during the summer of 2018 at Tuktoyaktuk than other locations in northern Canada".

Figure 6: Since the time series shown in Figure 5 do not indicate a strong diurnal signal, I wonder how representative is the averaged diurnal profiles for PM mass concentrations. It would be good to plot the mean, media and inter-quartile range to indicate variability. The largest diurnal variation seems to be in the 2 – 5 um range – do the author have any explanation?

Line 374 – 376: It is better to just state that the $PM_{2.5}$ levels observed at Tuktoyaktuk is well below the national air quality standard. I would suggest removing the latter part of the sentence "suggesting $PM_2$ likely had minimal effects on the air quality of the community".

4. Conclusion

Line 379 – 380: The authors stated that the analysis carried out could not identify distinct sources. Could the authors elaborate on the kind of information needed (or missing) for source identification? Simply stating that the site is influenced by a wide range of aerosol particle sources with complex processes seems overly general and nonspecific. What are the potential sources and processes influencing this site? It seems that the authors could delve into some of the available information (e.g., met and trajectory analysis) a bit more to gain some more insight into the observations at this Arctic site.

---

## Author Comment (AC1)

**Reviewer 1**

This study presents valuable atmospheric measurements from the remote areas of the Northern Territories in Canada. Tuktoyaktuk is a hamlet representing both continental and Arctic Ocean maritime climate conditions in summer, and thus important location to obtain information on atmospheric composition. The authors provide detailed analysis on the chemical composition of fine (PM2.5) and coarse (PM10-2.5) aerosol filter samples as well as PM mass concentrations from optical particle counter. Although they faced difficulties with the quantitative assessment of ions in metals in aerosol particle filter samples due to their low chemical mass, characteristic for such remote area in the Arctic, the report is important and valuable for further research of such areas. Available scientific literature was used to relate the results with similar Arctic locations. There is no clear conclusion what were the exact sources of captured aerosol particles, but the discussion explained in full extent the possible scenarios from where these aerosols could have originated.

The paper is clearly written and the figures represent well the analyses. I would recommend publication of this work after some minor revisions. Particularly, authors could add more information into the introduction, explain the methods/results on aerosol filter masses analysis in greater detail, focus possible local natural sources of aerosol – high latitude dust sources, biomass burning and marine aerosol, and correct the references not cited in the text.

**We thank the reviewer for their helpful comments. We have added information to the introduction, revised our paper to improve discussions of filter masses and sources of aerosol particles within the Arctic, and corrected references that were not cited in the original draft.**

Specific comments:

C1. L31-32, , L39-42 – consider to include the study of Boy et al. (2019) describing the role of aerosols in changing climate in the Arctic here.

Boy, M., Thomson, E. S., Acosta Navarro, J.-C., Arnalds, O., Batchvarova, E., Bäck, J., Berninger, F., Bilde, M., Brasseur, Z., Dagsson-Waldhauserova, P., Castarède, D., Dalirian, M., de Leeuw, G., Dragosics, M., Duplissy, E.-M., Duplissy, J., Ekman, A. M. L., Fang, K., Gallet, J.-C., Glasius, M., Gryning, S.-E., Grythe, H., Hansson, H.-C., Hansson, M., Isaksson, E., Iversen, T., Jonsdottir, I., Kasurinen, V., Kirkevåg, A., Korhola, A., Krejci, R., Kristjansson, J. E., Lappalainen, H. K., Lauri, A., Leppäranta, M., Lihavainen, H., Makkonen, R., Massling, A., Meinander, O., Nilsson, E. D., Olafsson, H., Pettersson, J. B. C., Prisle, N. L., Riipinen, I., Roldin, P., Ruppel, M., Salter, M., Sand, M., Seland, Ø., Seppä, H., Skov, H., Soares, J., Stohl, A., Ström, J., Svensson, J., Swietlicki, E., Tabakova, K., Thorsteinsson, T., Virkkula, A., Weyhenmeyer, G. A., Wu, Y., Zieger, P., and Kulmala, M., 2019.  Interactions between the atmosphere, cryosphere, and ecosystems at northern high latitudes, Atmos. Chem. Phys., 19, 2015-2061.

**R1. We have added this reference to the revised paper.**

C2. L37-38 – high latitude dust sources in the Arctic are active also in the winter. Please see examples here:

Mackay, J. R., & Burn, C. R. (2005). A long-termfield study (1951–2003) of ventifacts formed by kata-batic winds at Paulatuk, western Arctic coast, Canada. Canadian Journal of Earth Sciences,42(9), 1615–1635.

Bullard J.E., Baddock, M., Bradwell, T., Crusius, J., Darlington, E., Gaiero, D., Gassó, S., Gisladottir, G., Hodgkins, R., McCulloch, R., McKenna Neuman, Ch., Mockford, T., Stewart, H., Thorsteinsson, Th., 2016. High Latitude Dust in the Earth System. Reviews of Geophysics: DOI: 10.1002/2016RG000518.

Dagsson-Waldhauserova, P., Arnalds, O., Olafsson, H., 2014. Long-term variability of dust events in Iceland. Atmospheric Chemistry and Physics 14, 13411-13422. DOI:10.5194/acp-14-13411-2014.

Dagsson-Waldhauserova, P., Renard, J.-B., Olafsson, H., Vignelles, D., Berthet, G., Verdier, N., Duverger, V., 2019. Vertical distribution of aerosols in dust storms during the Arctic winter. Scientific Reports 6, 1-11.

**R2. We have added these references to the revised paper and added a statement about mineral dust as a source of aerosol particles during the winter-spring period in the Arctic:**

**Page 2, lines 36-38**

**"However, natural sources of aerosol particles, such as mineral dust, have been identified in the Arctic during the winter-spring period (Dagsson-Waldhauserova et al., 2014, 2019; Bullard et al., 2016; Mackay and Burn 2005)."**

C3. L66-67– Important aerosol-cloud climate feedback in the Arctic has been described by Murray et al. (2021) and Sanchez-Marroquin et al. (2020).

Murray, B. J., Carslaw, K. S., and Field, P. R.: Opinion: Cloud-phase climate feedback and the importance of ice-nucleating particles, Atmos. Chem. Phys., 21, 665–679, https://doi.org/10.5194/acp-21-665-2021, 2021.

Sanchez-Marroquin, A., Arnalds, O, Baustian-Dorsi, K., Browse, J., Dagsson-Waldhauserova, P., Harrison, A.D., Maters, E.C., Pringle, K.J., Vergara-Temprado, J., Burke, I.T., McQuaid, J.B., Carslaw, K.S., Murray, B.J., 2020. Iceland is an episodic source of atmospheric ice-nucleating particles relevant for mixed-phase clouds. Science Advances 6, eaba8137, 1-9.

**R3. We have added these references to the revised paper.**

C4. L165-167 – Could you please explain better why the **PM2.5** mean and maximum mass range from the filters **are higher than** for **PM10**? Are there other studies facing the same results? Consider to use also median.

**R4. It should be noted that the coarse mode reported here represents only the fraction of particles with aerodynamic diameters from 2.5 – 10 μm, and that the actual $PM_{10}$ is always higher than the $PM_{2.5}$, as expected. To clarify this, we now include the mean, median, and range of total coarse mode ($PM_{10}$) masses in the text.**

**Page 7, lines 184-194**

**"The fine ($PM_{2.5}$, mean ± SD, 15 ± 9 μg, median 15 μg), coarse only ($PM_{10-2.5}$, 14 ± 4 μg, median 14 μg), and total coarse ($PM_{10}$, 29 ± 10 μg, median 26 μg) aerosol filter masses were similar during the study period, with notable variability (Fig. 2). For instance, the masses range from 2.6-31 μg, 7.3-22 μg, and 17-44 μg in $PM_{2.5}$, $PM_{10-2.5}$, and $PM_{10}$, respectively. Snider et al. (2016) also reported that masses of $PM_{2.5}$ (median 72 μg, lower-upper quintiles 42-131 μg) and $PM_{10-2.5}$ (median 90 μg, lower-upper quintiles 44-154 μg) were comparable in filter samples collected across a global network of sites (i.e., Surface PARTiculate mAtter Network, SPARTAN) using an AirPhoton sampler, although the exact distribution is site-specific. For instance, comparable masses of $PM_{2.5}$ and $PM_{10-2.5}$ are not unexpected, considering coarse aerosol particle emissions are likely transient in nature (i.e., from local sources), and they may not have been sampled during the brief sampling period in this study. This is further supported by the mass distribution of fine and coarse aerosol particles measured by the particle counter, where the mass fraction of fine aerosol particles was occasionally higher than the mass fraction of coarse aerosol particles (Fig. S4). However, $PM_{10}$ masses in this study were always greater than $PM_{2.5}$ masses, as expected."**

C5. L208-217 – These elements are abundant in dust from the high latitude dust sources. Consider to add discussion also on that here. Are you aware of some active dust sources in your region? MacKenzie River, lakes, beaches? For example peninsula north of Paulatuk is an active source in July (as on the SDS map, but also in winter as published). You can find the Sand Dust Storm activity index at this website: https://maps.unccd.int/sds/ (choose July). Chemical composition of different HLD sources can be found here:

Crocchianti, S., Moroni, B., Dagsson-Waldhauserová, P., Becagli, S., Severi, M., Traversi, R., Cappelletti, D., 2021. Potential Source Contribution Function Analysis of High Latitude Dust Sources over the Arctic: Preliminary Results and Prospects. Atmosphere 12, 347-362.

Mackay, J. R., & Burn, C. R. (2005). A long-termfield study (1951–2003) of ventifacts formed by kata-batic winds at Paulatuk, western Arctic coast, Canada. Canadian Journal of Earth Sciences,42(9), 1615–1635.

**R5. We have discussed the possibility of other, high latitude dust source regions in the revised paper and included the recommended citations where appropriate.**

**Page 9, lines 242-246**

**"This could suggest that in addition to local sources, the presence of Al in filter samples was attributed to mineral dust emissions and atmospheric transport from other, high latitude regions within the Arctic (Crocchianti et al., 2021; Mackay and Burn 2005). For**

**example, it is possible that aerosol filter samples on 18 and 26 July (Fig. S1) were influenced by active dust sources near the North Slope of Alaska (https://maps.unccd.int/sds/).”**

C6. L235-240 – Biomass burning could be an explanation of your results here including high fluoride abundance. Consider to investigate if there was biomass burning event before or during your study period.

**R6. We agree that combustion activities could be a source of fluoride in our study. Unfortunately, we are unable to identify specific biomass burning event(s) before or during the study. It is difficult to assign a source to fluoride since it is derived from multiple sources in the environment (e.g., marine, mineral dust, and combustion aerosols). We discuss this challenge in the paper.**

C7. L477-479 – Jacobi et al. (2019) is not referred in the text

**R7. Thanks for pointing that out. We have removed this reference in the revised paper.**

C8. L538-539 – Thakur and Thamban (2019) is not referred in the text

**R8. Thanks for pointing that out. We have removed this reference in the revised paper.**

C9. L555-557 – Willis et al. (2019) is not referred in the text

**R9. Thanks for pointing that out. We have removed this reference in the revised paper.**

---

## Author Comment (AC2)

**Reviewer 2**

The manuscript presents a report on the measurements carried out at a small Inuit community located near the Mackenzie River delta during summertime in 2018. The measurements include filter samples and continuous sampling using a particle counter. Analysis were carried out in an effort to characterise particulate matter compositions observed at this sites during the summer period. The manuscript is written as a measurement report, due to, I gather, some issues with the measurements (e.g., missing sample volume information, insufficient sampling period due to low concentrations) which resulted in certain limitations of the study. I do feel that the authors could extend the analysis a bit more to improve the interpretation of the measurement results and gain better insight into the observation at this remote Arctic site. My comments and suggestions are provided below.

**We thank the reviewer for their helpful comments. We have expanded our analysis and interpretations in the revised paper.**

2. Method and data analysis

2.1 Study area and sample collection

C1. Could the authors provide some background on why this particular site was chosen and a description of the site and the area in terms of potential sources of aerosols?

**R1. As the reviewer is likely aware, working in northern communities requires good relationships with community members and we were allowed access to this particular site based on of our past collaborations. The town of Tuktoyaktuk was chosen because the larger research objective of the study was to characterize polar fog in northern coastal communities, which is not within the scope of this manuscript and for which the analysis is still ongoing. The town had also just been connected by road to Inuvik, meaning it was experiencing increasing road traffic, which we thought would be interesting to characterize. We have included more information about the site and possible sources in the revised paper:**

**Pages 4-5, lines 111-118**

**"The ACCC is located near residential and institutional buildings, a beach, and an unpaved road (all < 100 m from the site). Potential local sources of aerosol particles at the site may include dust (e.g., beach sand, road and mineral particles), as well as marine and combustion aerosol particles resulting from natural and anthropogenic activities (e.g., sea spray and vehicle traffic). The ACCC could also be influenced by regional sources in northern Canada and the United States. For example, emissions from the ignition of lignite in the Smoking Hills and a migratory bird colony on Banks Island are possible sources of natural aerosol particles and gases, whereas emissions from the Prudhoe Bay Oil Field could be an anthropogenic source of aerosol particles and gases (Fig. 1). This site was selected for this study to represent a northern coastal community that was undergoing increased human activities (due to recent road access via the Trans-Canada Highway)."**

C2. The authors acknowledge that the sample volume could not be determined due to a file writing error so that air concentrations cannot be discerned from the filter samples. Were airflow rate and length of sampling controlled?

**R2. According to available instrument log files, the airflow rate was maintained at 4.4 L/min, and the sampling time was three hours (0600 to 0900 local time). Unfortunately, we are not able to confirm the sampling flow rate and time for all of the filters due to the file writing error (i.e., there are missing instrument log files). We elected to omit the airflow rate because we are only reporting total mass.**

C3. Is it correct that the particle counter has a lower size cut at 300 nm (so that particles smaller than 300nm are not measured by this instrument)? This seems to limit the ability to characterize aerosol size spectrum from this study as it misses Aitken mode almost entirely.

**R3. Yes, that is correct. The instrument is more useful in understanding aerosol mass distributions, which was the focus of our study. We have noted this focus in the revised paper:**

**Page 4, lines 99-101**

**"This unit contains six channels that simultaneously measures aerosol particles binned by diameter with lower limits of 0.3, 0.5, 1, 2, 5, and 10 μm, allowing the aerosol mass distribution to be characterized."**

C4. Were the inlets (filter sampling and particle counter) collocated?

**R4. Yes, these instruments were collocated. This is clarified in the revised paper:**

**Page 4, lines 102-103**

**"The instruments in this study were collocated and mounted approximately 3.5 m above ground level at the Aurora College Community Centre (ACCC) in Tuktoyaktuk."**

2.3 QA/QC

C5. Line 121: "the unanticipated, brief sampling period" – what do you mean?

**R5. Before deploying the instrument, we had planned for each filter to sample ambient aerosol particles for three hours per day over the course of one week (i.e., 21 total hours of sampling per filter). Unfortunately, due to a file writing error, each filter appears to have sampled for only three hours in total. This is what we were referring to as "unanticipated". We have removed this word to improve clarity in the revised paper.**

C6. Line 121 – 124: How should this low mass issue be taken into consideration with regard to the results shown in Figure 2?

**R6. The magnitude of chemical mass reported in our study may carry uncertainty since we are extrapolating masses outside of our quantitation range. We would have to reanalyze the samples at a lower concentration range to confirm the magnitude of chemical masses, but unfortunately, the samples have since been discarded. Although the magnitude of chemical mass may carry some uncertainty, the intention of this dataset is to provide an initial assessment of the chemical composition of aerosol particles for a region of the Canadian Arctic in which there are limited measurements.**

**We have added a similar statement to the QA/QC section in the revised paper:**

**Page 6, lines 146-148**

**"Although the magnitude of chemical mass reported in this study may carry uncertainty, the intention of this dataset is to provide a first assessment of the chemical composition of aerosol particles in a relatively underreported region of the Canadian Arctic."**

2.4 Data analysis

C7. Although the authors did include statistical summaries of meteorological observations (from the Tuktoyaktuk airport) during the field measurement period (Figure S1), it would be much more useful for data interpretation to plot the time series (e.g., wind speed/direction, temperature, humidity) as well.

**R7. We have included a new figure showing time series for wind speed and direction, temperature, and relative humidity during the estimated aerosol filter sampling periods (0600 to 0900 local time) in the supplement and used these data to improve our interpretations in the revised paper.**

**Page 7, lines 178-180**

**"Meteorological data, including temperature, wind speed and direction, and relative humidity was retrieved from historical climate archives (Environment and Climate Change Canada, 2020) for the Tuktoyaktuk airport during the study (Fig. S3)."**

3. Results and discussion

3.1 Aerosol filter masses

C8. Line 167 – 169: The authors seems to suggest that the $PM_{2.5}$ and $PM_{10\text{-}2.5}$ masses based on filter measurements at this site were comparable (in terms of means and ranges; what about median?) and that the comparable masses between $PM_{2.5}$ and $PM_{10\text{-}2.5}$ were also shown from the global SPARTAN network sites (using the same instrument and analysis method). What does this imply? Is this corroborated by the mass estimates based on the particle counter measurements?

**R8. We also find that median masses are comparable between fractions, which are now included in the revised paper. The comparison with the SPARTAN network was meant to**

imply that it is not unusual for the two size fractions to be similar, since neither our measurements nor the SPARTAN measurements consistently show one fraction to be higher than the other. In fact, the breakdown between the two modes is site-specific. For example, figures 8.11, 8.13, 8.18 and 8.19 from Seinfeld and Pandis (Atmospheric Chemistry and Physics, 2nd Edition, John Wiley & Sons, Inc., New Jersey, 2006) show volume concentration, which is proportional to mass concentration, of particles smaller than 2.5 μm to be comparable to, if not exceeding the volume concentration of particles larger than 2.5 μm in urban, next to a freeway, remote continental, and free tropospheric measurement locations, respectively. In addition, it is expected that the coarse mode particles are at least sometimes generated through episodic occurrences, which may not always be captured during our sample times since we suspect that each filter only sampled for approximately three hours.

As suggested by the reviewer, we also calculated the mass fraction of the fine and coarse mode aerosols based on the optical particle counter, which showed that the mass fractions between the two modes were comparable to the filter samples. We selected 26 July and 3, 11, and 19 August for this comparison because there are available log files indicating the sampling time. These details have now been included in the text.

Page 7, lines 184-194

"The fine ($PM_{2.5}$, mean ± SD, 15 ± 9 μg, median 15 μg), coarse only ($PM_{10-2.5}$, 14 ± 4 μg, median 14 μg), and total coarse ($PM_{10}$, 29 ± 10 μg, median 26 μg) aerosol filter masses were similar during the study period, with notable variability (Fig. 2). For instance, the masses range from 2.6-31 μg, 7.3-22 μg, and 17-44 μg in $PM_{2.5}$, $PM_{10-2.5}$, and $PM_{10}$, respectively. Snider et al. (2016) also reported that masses of $PM_{2.5}$ (median 72 μg, lower-upper quintiles 42-131 μg) and $PM_{10-2.5}$ (median 90 μg, lower-upper quintiles 44-154 μg) were comparable in filter samples collected across a global network of sites (i.e., Surface PARTiculate mAtter Network, SPARTAN) using an AirPhoton sampler, although the exact distribution is site-specific. For instance, comparable masses of $PM_{2.5}$ and $PM_{10-2.5}$ are not unexpected, considering coarse aerosol particle emissions are likely transient in nature (i.e., from local sources), and they may not have been sampled during the brief sampling period in this study. This is further supported by the mass distribution of fine and coarse aerosol particles measured by the particle counter, where the mass fraction of fine aerosol particles was occasionally higher than the mass fraction of coarse aerosol particles (Fig. S4). However, $PM_{10}$ masses in this study were always greater than $PM_{2.5}$ masses, as expected."

C9. Line 169 – 171: Could the authors elaborate on this a bit more? How are the meteorological conditions related to the observed PM mass levels and how are PM levels affected by local and distant sources?

R9. We have provided clarification in the revised paper:

Page 7, lines 195-200

"It is expected that the size distribution of aerosol particles (e.g., coarse mineral dust vs. fine combustion aerosol particles) and local meteorology could affect the magnitude of filter masses during the study. For example, it is possible that warmer temperatures during 26 July (Fig. S3) may have enhanced local emissions of coarse aerosol particles through heating and convection, contributing to a high $PM_{10-2.5}$ mass, while precipitation (i.e., drizzle, rain, and snow), which was observed at the airport before and during sampling events (Fig. S5), could have reduced atmospheric loads through the action of scavenging aerosol particles and gases."

3.2 Chemical composition of aerosol filters

C10. Figure 2 shows both the gravimetric masses and chemical masses from each of the filter samples. It would be interesting to see the mass differentials between the gravimetric mass and the total chemical mass from each of the samples to get an idea on how much of the PM mass is explained by the speciation and how much is unexplained (given that the analysis covers inorganic ions and metals but not organics). Perhaps this will provide some additional information for source identification under different conditions.

R10. We have included a new figure in the supplement showing the contribution of chemical mass to the gravimetric mass. However, it is challenging to use this information for source identification since other species (organic carbon, black carbon, other inorganic ions, etc.) were not measured. We discuss this limitation in the paper:

Page 13, lines 340-347

"In addition to uncertainties associated with the unclassified inorganic fraction, a substantial fraction of mass on aerosol filter samples could not be identified. For instance, the average chemical mass only accounted for 10 % and 12 % of the total gravimetric mass in $PM_{2.5}$ and $PM_{10-2.5}$ filter samples, respectively (Fig. S6). Discrepancies between the total gravimetric mass and chemical mass of aerosol filter samples could be attributed to analytical uncertainties, the loss or gain of volatile species from filters after sampling (Saltzman, 2013), and/or contributions from untargeted chemical components. For example, it is possible that other inorganic ions and metals, organic material, and black carbon were components of aerosol particles at Tuktoyaktuk, pursuant to the chemical composition of aerosol particles in other Arctic regions (Kadko et al., 2016; Leaitch et al., 2018; Conca et al., 2019; Ferrero et al., 2019; Sharma et al., 2019)."

C11. Is there any correlation between the variation in gravimetric masses amongst the filter samples and the variation in PM concentrations derived from the aerosol counter measurement?

R11. While this a good suggestion for comparing the collocated instruments, we are unable to confirm the sampling times for most of the filters, limiting direct comparisons between the instruments. As such, the representativeness of the correlation analysis results would be unknown, carrying large uncertainties. However, we found that the mass distributions are consistent with the filter analysis, as per the reviewer's suggestion (see R8 and Fig. S4).

C12. Is sodium not analysed? Is sulfate shown including sea-salt sulfate?

**R12. Sodium was a targeted cation in our analysis, but it was found in high concentrations in the field and laboratory blanks, and as a result, it could not be reliably quantified in the filter sample extracts. The sulfate shown is the total concentration (i.e., sea salt and non-sea salt sulfate).**

C13. Line 193 – 195: It might be good to rephrase this, as the only common feature shared in chemical composition of metals between Tuktoyaktuk and other Arctic sites shown in Figure 3 is the dominance of Al and Fe.

**R13. We have rephrased this statement in the revised paper:**

**Page 9, lines 222-223**

**"Similar to other Arctic regions (Fig. 3), Al and Fe dominated aerosol filter samples at Tuktoyaktuk, which have been linked to mineral dust emissions (Liberda et al., 2015; Ferrero et al., 2019)."**

C14. Figure 3: Are chemical composition profiles from other Arctic sites shown here based on summertime measurements also? If not, how might seasonal variability affect the comparison here? Also for the comparison do all sites carry out analysis for the same suite of ions and metals? For example, the Tuktoyaktuk profiles do not include sodium; does it mean that sodium is not present or just not analysed?

**R14. Most of the data from the other Arctic sites were collected during the summer, with the exception of Landsberger et al. (winter) and Conca et al. (spring/summer). We have also noted that seasonal variability and the analytes targeted in a study are important factors when comparing profiles across sites in the revised paper:**

**Page 9, lines 225-230**

**"It is important to note that metal profiles in Landsberger et al. (1990) and Conca et al. (2019) are based on data collected during winter/spring periods, therefore seasonal differences in aerosol particles source in those studies may account for differences in composition profiles in comparison to Tuktoyaktuk (e.g., Arctic haze versus summertime sources). In addition, the studies compared in Fig. 3 do not always target the same ions and metals and/or face analytical challenges preventing accurate reporting of data, which collectively could also contribute to chemical composition differences across sites."**

**Ions and metals that were below the detection limit in our study are not included in Fig 3. We have provided clarification in its caption:**

**"Only ions and metals that are equal to or greater than the detection limit in this study are included in this figure."**

C15. Line 211 – 214: Longer back trajectories are needed to better discern air mass origin (or influence) in the Arctic summertime, given that air mass tends to resides within the Arctic region for a long time (up to 2 weeks) in summertime (Stohl, 2006, JGR).

**R15. We present 5-day back trajectories in the revised paper. Note that we have changed the meteorology input data (Global Data Assimilation System, GDAS) to accommodate longer trajectories since there are missing NARR meteorology data (e.g., maximum back trajectory duration is 2 days for particular days). We have used these longer trajectories to improve discussions of aerosol particle sources.**

**Page 6, lines 166-168**

**"Air mass back trajectories were calculated over 120 hours using Global Data Assimilation System (GDAS) meteorology, setting the heights at the same location (end of parcel trajectory) to be 50, 200, and 400 m above ground level (Figs. S1 and S2)."**

C16. Line 235 – 240: Do the author imply that the $Cl^-$ and $Br^-$ detected in $PM_{2.5}$ and $PM_{10-2.5}$ samples, respectively could be of biomass burning origin? It would have been possible for $Cl^-$ in $PM_{2.5}$ but one would not expect coarse particles to be transported from a long distance. It is still surprising not to see $PM_{10-2.5}$ sea salt at this coastal site.

**R16. Yes, that is what is implied. We agree that this is a surprising result since this is a coastal site. However, local wind speeds were often low and did not always originate from coastal areas during several sampling periods (see Fig. S3). This is discussed in the revised paper:**

**Page 10, lines 257-262**

**"It is interesting that $Cl^-$ was only detected in $PM_{2.5}$ while $Br^-$ was only detected in $PM_{10-2.5}$ filter samples since $Cl^-$ and $Br^-$ have been measured in seawater from the Canadian Arctic Archipelago (Xu et al., 2016). Although the ACCC is a costal site, surface meteorology records from the airport (Fig. S3) indicated that local wind speeds were often below 4 m s$^{-1}$ (e.g., 26 July, 19 and 27 August, 12 September), which has been suggested as a threshold wind speed for whitecap formation (O'Dowd and de Leeuw, 2007). However, a marine influence was expected during 3 and 11 August and 4 September, since wind speeds were greater than 4 m s$^{-1}$ and originated from north westerly and easterly directions (Fig. S3)."**

**We also discuss the possibility that organic acids, which were not measured in this study, could also contribute to chloride depletion in aerosol particles:**

**Pages 12, lines 310-312**

**"Although the organic composition of aerosol filter samples was not characterized in this work, it is important to consider that organic acids (Laskin et al., 2012) may have also contributed to $Cl^-$ depletion in aerosol filter samples."**

C17. It would be helpful to include a description of the local and regional sources (natural and anthropogenic). The influence of Smoking Hills emissions and Prudhoe oil fields could be discerned from trajectory analysis. For example, the August 3 sample could be influenced by sulfur emissions from Smoking Hills (based on the trajectory shown in Figure S2).

**R17. We have added a description of local and regional sources in Section 2.1, and used the air mass trajectories to improve our discussion of metal and ion sources:**

**Pages 9-10, lines 250-256**

**"Other air mass trajectories originating from west and north westerly directions (i.e., Alaska and Russia) were observed on 18, 26 July and 19 August (Figs. S1 and S2), and filter samples during these periods contained Ba, Ag, and Sb. It is possible that these air masses were influenced by emissions from the Prudhoe Bay Oil Field and mining activities in Alaska, Russia, and Canada during these periods (Alaska Miners Association, 2020; Government of Canada, 2018; European Environment Agency, 2017). However, local emissions from combustion and natural or anthropogenic dust (e.g., road dust containing tire wear and mineral/soil particles) (Snider et al., 2016; Crocchianti et al., 2021; Mackay and Burn 2005) cannot be precluded as sources of Al, Fe, Ti, Zn, Ba, Ag, and Sb in filter samples."**

**Page 11, lines 297-306**

**"In addition, sulfur emissions from the ignition of lignite in the Smoking Hills (Radke and Hobbs, 1989) was a likely natural source of $SO_4^{2-}$ in $PM_{2.5}$ at Tuktoyaktuk, especially on 3 and 27 August, according to air mass back trajectories (Figs. S1 and S2). An additional source of $SO_4^{2-}$ in $PM_{2.5}$ at Tuktoyaktuk may include anthropogenic emissions from the combustion of fossil fuels (e.g., vehicles, aircraft, boats, etc.) (Leaitch et al., 2018; Willis et al., 2018). Other ions characteristic of combustion were also identified in aerosol filter samples from Tuktoyaktuk, such as $NO_3^-$ and $NH_4^+$, possibly from the emission and oxidation of nitrogen oxides ($NO_x$) and emissions of ammonia during fossil fuel combustion from local and long-range sources. However, ammonia emissions in the Arctic have also been associated with natural sources, such as soil (Wentworth et al., 2016) and guano (Croft et al., 2016; Wentworth et al., 2016), which could account for $NH_4^+$ in aerosol filter samples at Tuktoyaktuk, particularly on 3 August because $NH_4^+$ was detected in air masses that travelled near a bird colony on Banks Island before arriving at the ACCC (Fig. S1)."**

3.3 Size distribution, temporal variability, and health implications of aerosol particles

C18. It should be noted that the aerosol number size distribution based on this measurement is incomplete as the measurement is missing Aitken mode particles almost entirely (with the lowest size cut at 300 nm).

**R18. We have noted this in Section 3.3 by adding the phrase "of particles larger than 0.3 μm" where appropriate.**

C19. Line 318: What do you mean by number size distribution being consistent with Herenz et al. (2018)? Their number size distributions show highest mode at ~40 – 50 nm under polluted conditions and just below 200 nm under clean conditions (their Figure 5). Those measurements were conducted during spring-to-summer transition period while this study is during summer period. One would expect to see quite significant differences in aerosol size distribution and chemical composition between these two different periods. Would this not be the case?

**R19. We agree this is not a clear comparison as written and have removed it in the revised paper:**

**Page 14, lines 364-366**

**"The average number size distributions of particles larger than 0.3 µm were similar throughout the study, with particle number concentrations highest in the 0.3-0.5 µm bin (Fig. S7). The mass size distributions also remained similar throughout the study, with mass concentrations dominated by the 2-5 µm aerosol particles (Fig. S7)."**

C20. Table 1: Please clarify on $PM_{2.5}$ and $PM_{10}$ measurements at the NAPS sites. They may be using different instrument/technique than that used in this study.

**R20. We have discussed that possibility in Section 3.3:**

**Page 15, lines 390-392**

**"For example, one method used by the National Air Pollutant Surveillance program to determine mass concentrations is by filtration and beta attenuation (Canadian Council of Ministers of the Environment, 2019) whereas the method used here relies on aerosol particle number concentrations and estimations of aerosol particle density (Eq. 1)."**

C21. Line 326, Line 339, and Line 343: It may be more appropriate not to use the term "discrepancy" (or "discrepancies") here. The differences are expected between these different northern sites, due to, as the authors pointed out, the differences in geographical locations, local and regional sources, etc.

**R21. We have replaced "discrepancy" with "difference" where appropriate.**

C22. Line 343 – 344: It may be better to say "… concentrations were lower during the summer of 2018 at Tuktoyaktuk than other locations in northern Canada".

**R22. We have made the suggested change.**

C23. Figure 6: Since the time series shown in Figure 5 do not indicate a strong diurnal signal, I wonder how representative is the averaged diurnal profiles for PM mass concentrations. It would be good to plot the mean, media and inter-quartile range to indicate variability. The largest diurnal variation seems to be in the 2 – 5 um range – do the author have any explanation?

**R23. The new figure is included in the revised paper, with an updated discussion:**

**Page 17, lines 415-418**

**"Aerosol particle mass concentrations did not exhibit notable diurnality during the study (Fig. 6). Average mass concentrations were typically higher than median mass concentrations and exhibited notable variability in the 2-5 and 5-10 μm size bins, which are likely driven by enhanced aerosol particle emissions from local human activities at Tuktoyaktuk, as discussed previously (i.e., festival and weekend activities)."**

C24. Line 374 – 376: It is better to just state that the $PM_{2.5}$ levels observed at Tuktoyaktuk is well below the national air quality standard. I would suggest removing the latter part of the sentence "suggesting $PM_2$ likely had minimal effects on the air quality of the community".

**R24. We have made the recommended change in the revised paper.**

4. Conclusion

C25. Line 379 – 380: The authors stated that the analysis carried out could not identify distinct sources. Could the authors elaborate on the kind of information needed (or missing) for source identification? Simply stating that the site is influenced by a wide range of aerosol particle sources with complex processes seems overly general and nonspecific. What are the potential sources and processes influencing this site? It seems that the authors could delve into some of the available information (e.g., met and trajectory analysis) a bit more to gain some more insight into the observations at this Arctic site.

**R25. We have clarified possible sources and processes at our site and information that could be beneficial in future work for source identification:**

**Page 17, lines 424-443**

**"The chemical composition of aerosol filter samples and concentration of aerosol particles from Tuktoyaktuk were determined during July-September 2018. Although our analysis could not identify distinct sources, the results suggest that this moderately-sized community in the Canadian north was influenced by a wide range of aerosol particle sources with complex processes. The observed aerosol particles were likely derived from local natural sources like marine and mineral dust and anthropogenic sources like the combustion of fossil fuels and road dust, while emissions from the Prudhoe Oil Field, Smoking Hills, bird colonies on Banks Island, mining activities in northern Canada, Russia, and Alaska, and mineral dust from active source regions in the Arctic are possible regional sources of aerosol particles, pursuant to air mass back trajectory analysis (Figs. S1 and S2). We hypothesize that precipitation reduced atmospheric loads of aerosol particles and gases during the study, which is expected to affect the magnitude of the gravimetric mass and chemical composition of aerosol filters and at Tuktoyaktuk, and air temperature may have enhanced local emissions of coarse aerosol particles through daytime heating and convection. Our analysis indicates that there were significant, unknown components**

identified in aerosol filter samples during the summer of 2018 at Tuktoyaktuk, which may influence the atmospheric fate of aerosol particles in the Arctic troposphere. While the mass concentrations of $PM_2$ were found to be significantly lower at Tuktoyaktuk compared to the Canadian Ambient Air Quality Standard, it is likely that their concentrations will increase in the future due to climate change, which is expected to promote increases in ship and air traffic in the Arctic as well as the number of ice-free days and natural emissions from open waters. Although these measurements only represent a snapshot of the aerosol particles at Tuktoyaktuk, they can nevertheless provide insights into the chemistry and concentration of aerosol particle samples, which can be used in the future to assess aerosol particle chemistry and air quality in the Canadian Arctic. Future work should focus on constraining possible sources of aerosol particles, such as acquiring time-resolved chemical mass spectra data and performing factor analysis (e.g., positive matrix factorization) and/or analysing the chemical composition of local soils."

---

## Referee Report (RR1)

**Reviewer's report on the revised manuscript by MacInnis et al. "Measurement report: The chemical composition and temporal variability of aerosol particles at Tuktoyaktuk, Canada during the Year of Polar Prediction Special Observing Period", Atmospheric Chemistry and Physics, Manuscript ID: acp-2021-262**

In the revised manuscript, the authors have addresses most of my previous comments/concerns. The only significant change that I would suggest is to replace the plots in Figure S3 with continuous time series of meteorological observations (e.g., temperature, relative humidity, surface pressure, wind speed/direction) for the field study period, i.e., July 18 – Sept. 12, instead of during filter sampling periods only. The continuous time series is more useful for providing synoptic context and discerning different air masses influencing the measurements. It can also relate better to the continuous PM mass observations shown in Figure 5.

It is perhaps worth noticing that July 26 could be a good case when the site may be influenced by the Prudhoe Bay Oil Fields emissions, as the back-trajectories, particularly the low-level one, shown in Figure S1, circling around the oil field before arriving at the measurement site. The fact that the water-soluble ions are dominated by sulfate and nitrate may also corroborate this suggestion. Another case of possible interest may be August 27 – the 5-day back-trajectories suggest possible influence of biomass burning pollutants from northern Canada. Does the chemical mass analysis show any indication of biomass burning influence for this day?

---

## Author Response (AR2)

Reviewer Comments

C1. In the revised manuscript, the authors have addresses most of my previous comments/concerns. The only significant change that I would suggest is to replace the plots in Figure S3 with continuous time series of meteorological observations (e.g., temperature, relative humidity, surface pressure, wind speed/direction) for the field study period, i.e., July 18 – Sept. 12, instead of during filter sampling periods only. The continuous time series is more useful for providing synoptic context and discerning different air masses influencing the measurements. It can also relate better to the continuous PM mass observations shown in Figure 5.

R1. Continuous time series for the suggested variables are provided in the Supplement. However, after examining the continuous time series, we find that they do not improve our understanding of the air masses influencing filter measurements. We believe our discussion provides a comprehensive assessment of the meteorology and air masses affecting filter samples during our study.

C2. It is perhaps worth noticing that July 26 could be a good case when the site may be influenced by the Prudhoe Bay Oil Fields emissions, as the back-trajectories, particularly the low-level one, shown in Figure S1, circling around the oil field before arriving at the measurement site. The fact that the water-soluble ions are dominated by sulfate and nitrate may also corroborate this suggestion. Another case of possible interest may be August 27 – the 5-day back-trajectories suggest possible influence of biomass burning pollutants from northern Canada. Does the chemical mass analysis show any indication of biomass burning influence for this day?

R2. We agree it is likely that emissions from the Prudhoe Oil Field and biomass burning were sources of ions and metals at our site. However, identifying these sources is challenging since many ions detected in the filters are from several sources in the environment (e.g., sulphate is from anthropogenic and natural sources). Furthermore, identifying biomass burning sources is also challenging because we did not detect potassium in our samples (a common tracer for biomass burning) and other tracers (e.g., levoglucosan) were not targeted. Nevertheless, we discuss these possibilities in the revised paper.

**Page 11, lines 296–305**

"Lower molar ratios of $Cl^-/SO_4^{2-}$ in aerosol filter samples could be attributed to non-oceanic sources of $SO_4^{2-}$ (i.e., natural and anthropogenic combustion sources). For example, $SO_4^{2-}$ in $PM_{2.5}$ at Tuktoyaktuk may have originated from natural sources, such as the biogenic emission and subsequent oxidation of dimethyl sulphide from the ocean (Bates et al., 1987). In addition, sulfur emissions from **the Prudhoe Oil Fields and the** ignition of lignite in the Smoking Hills (Radke and Hobbs, 1989) **were** likely **sources** of $SO_4^{2-}$ in $PM_{2.5}$ at Tuktoyaktuk, according to air mass back trajectories (Figs. S1 and S2). **Another** source of $SO_4^{2-}$ in $PM_{2.5}$ at Tuktoyaktuk may include anthropogenic emissions from the combustion of fossil fuels (e.g., vehicles, aircraft, boats, etc.) (Leaitch et al., 2018; Willis et al., 2018). Other ions characteristic of combustion were also identified in aerosol filter samples from Tuktoyaktuk, such as $NO_3^-$ and $NH_4^+$, possibly from the emission and oxidation of nitrogen oxides ($NO_x$) and emissions of ammonia during fossil fuel combustion from local, **regional (e.g., Prudhoe Oil Fields),** and long-range sources."

**Page 11, lines 281–286**

"In addition to marine sources, $Cl^-$ and $Br^-$ can originate from biomass burning. **Keene et al. (2006) identified** hydrochloric acid (HCl), chlorine ($Cl_2$), hypochlorous acid (HOCl), bromine ($Br_2$), and hypobromous acid (HOBr) as products of biomass burning, which could have been the

source of either Cl⁻ measured in the fine mode or Br⁻ measured in the coarse mode at Tuktoyaktuk. **While we are unable to confirm this source in our study, it is conceivable that biomass burning in northern Canada was a possible source of aerosol particles at Tuktoyaktuk (e.g., 27 August, Fig. S2)."**